# Riverine nitrogen supply to the global ocean and its limited impact on global marine primary production: a feedback study using an Earth System Model

Miriam Tivig[1,2], David P. Keller[1], and Andreas Oschlies[1]

[1]GEOMAR Helmholtz-Zentrum für Ozeanforschung Kiel, Düsternbrooker Weg 20, 24105 Kiel, Germany
[2]Deutscher Wetterdienst, Abteilung Klima und Umwelt, Güterfelder Damm 87-91, 14532 Stahnsdorf, Germany.

**Correspondence:** Miriam Tivig (mtivig@geomar.de)

**Abstract.** A common notion is that negative feedbacks stabilize the natural marine nitrogen inventory. Recent modeling studies have shown, however, some potential for localized positive feedbacks leading to substantial nitrogen losses, in regions where nitrogen fixation and denitrification occur in proximity to each other. Here we include dissolved nitrogen from river discharge in a global 3-D ocean biogeochemistry model and study the effects on near-coastal and remote open-ocean biogeochemistry.
We find that at steady state the biogeochemical feedbacks in the marine nitrogen cycle, nitrogen input from biological $N_2$ fixation, and nitrogen loss via denitrification, mostly compensate for the imposed yearly addition of 22.8 to 45.6 Tg of riverine nitrogen and limit the impact on global marine productivity to $< 2$ %. Global experiments that regionally isolate river nutrient input show that sign and strength of the feedbacks depend on the location of the river discharge and the oxygen status of the receiving marine environment. Marine productivity generally increases in proximity to the nitrogen input, but we also find a decline in productivity in the modelled Bay of Bengal and near the mouth of the Amazon River. While most of the changes are located in shelf and near coastal oceans, nitrogen supply from the rivers can impact the open ocean, due to feedbacks or knock-on effects.

*Copyright statement.* TEXT

## 1 Introduction

Nitrogen plays a key role in marine biogeochemistry in coastal and open oceans, as it is one of the major limiting nutrients for algal photosynthesis. Variations in the oceanic fixed-nitrogen (N) inventory are known to have driven marine productivity changes contributing to atmospheric $CO_2$ variations in Earth's history (Falkowski, 1997).

Although several studies questioned the stability of the global N budget (Codispoti et al., 2001; Gruber and Sarmiento, 1997; Codispoti, 1995), the present marine N inventory is generally considered to be in steady state (Deutsch et al., 2007; Altabet, 2006; Gruber, 2004; Tyrrell, 1999; Redfield et al., 1963). Oceanic fixed nitrogen concentrations are mainly controlled by the balance between denitrification and N fixation, creating negative feedbacks which damp the often strong pertubations of the

marine N content with respect to the more slowly overturning phosphorus (P) inventory (Somes et al., 2013; Deutsch et al., 2007; Gruber, 2004; Ruttenberg, 2003).

In regions where fixed N is sparse, organisms that fix atmospheric nitrogen ($N_2$) dissolved in seawater, commonly called di-
azotrophs, can compensate N deficits (Deutsch et al., 2007, 2001; Capone et al., 1997). However, as diazotrophs consume phosphate while adding fixed N to their environment, they are generally considered to regulate their own population. Previous studies have shown, that at least the most commonly cultured diazotrophs, especially Trichodesmium, have low growth rates relative to many non-fixing phytoplankton (e.g. Capone et al. (1997)). Indeed, while these organisms are able to fix N2 when reactive nitrogen is scarce, this turns into an disadvantage in regions where N is more abundant, because N2-fixation requires
more energy (Tyrrell, 1999). According to common conceptual models of controls on nitrogen fixation, the slowly growing diazotrophs are then out-competed by non-fixing phytoplankton, if enough P and other nutrients are present (Tyrrell, 1999). Note that part of our knowledge of $N_2$-fixation and most modern model concepts are still based on the original limited assumptions based on a few species, especially Trichodesmium. Also, emphasis has traditionally been put on bottom-up controls, despite accumulating evidence that top-down controls such as selective grazing on dominant species may have substantial impacts on
the distribution of diazotrophs and nitrogen fixation (Landolfi et al., 2021).

Denitrification is a metabolic process in which nitrate ($NO_3$) replaces oxygen ($O_2$) as terminal electron acceptor for respiration and is reduced to $N_2$, which is not bioavailable for most marine organisms, except diazotrophs (Gruber, 2004; Deutsch et al., 2001). Denitrification represents the main sink for fixed N in the ocean and occurs both in marine sediments and in the water columns under suboxic conditions (Gruber, 2004; Codispoti et al., 2001). But denitrification limits itself by reducing the con-
centrations of fixed N at the surface, which in turn limits the growth of phytoplankton and the heterotrophic $O_2$ consumption during organic matter remineralization, eventually making $NO_3$ less competitive as electron acceptor, where $O_2$ remains available (Landolfi et al., 2013; Gruber, 2004). These two processes, $N_2$ fixation and denitrification, both contribute to regulating the global marine N budget.

Beside the fixation of atmospheric $N_2$, rivers are also a major source of N to the coastal and the open ocean. Rivers are esti-
mated to add 36-60 Tg N $yr^{-1}$ to the coastal waters (Beusen et al., 2016; Mayorga et al., 2010; Seitzinger et al., 2005). These N inputs are regionally highly diverse and range over several orders of magnitude (Meybeck et al., 2006).

Although riverine N is not the main source of N to the marine environments, it can become a key player, as it is directly influenced by human activities. Seitzinger et al. (2010) e.g. estimated that global nitrogen export by rivers to the coastal waters increased by 17.7 % from 1970 to 2000. Nitrogen is known to impact coastal marine biology and biogeochemistry, leading
for example to eutrophication, algal blooms or hypoxia (e.g. Seitzinger et al., 2010; Billen and Garnier, 2007; Smith et al., 2003). Previous studies have shown, that nutrient input from land also has consequences for sea water composition and by this impacts biogeochemical processes in the open ocean farer away from the coasts (e.g. Barron and Duarte, 2015; Bauer et al., 2013; Bernard et al., 2011; Jahnke, 2010).

At this place it is appropriate to include some remarks about the anthropogenic perturbation of the N-cycle. In the Anthro-
pocene, human activities have led to increased inputs of fixed nitrogen from the atmosphere and through different sources of runoff from land (e.g. Somes et al., 2016; Kim et al., 2014, Lamarque et al.,2013). At the same time, warming and deoxygena-

tion can lead to increased N loss (Oschlies et al., 2019). The question has been therefore raised, if the global N budget could still be considered to be in steady state. While these combined effects on the N budget are still very uncertain, some studies suggest, that imbalances could be limited due to internal feedbacks of the N cycle (Landolfi et al., 2017; Somes et al., 2016; Krishnamurthy et al., 2006).

As global measurements of N concentrations and fluxes are difficult, models are often used to study the marine N cycle and its feedbacks. However, often global biogeochemical ocean models still omit riverine nutrient input to the ocean or represent it in a very simplified form (Séférian et al., 2020). Giraud et al. (2008) for example, tested the sensitivity of the global ocean biogeochemistry to coastal nutrient fluxes in a global ocean biogeochemistry model by introducing nutrients in different scenarios in the coastal grid boxes. They found that excess nutrients in the coastal ocean could impact the biological activity not only locally but also in the open ocean and that the effect depended more on the ratio between these nutrients and iron and silicate, than on the actual quantities. Nevertheless, the study by Giraud et al. (2008) was an idealized experiment without observed nutrient fluxes and a relatively simple representation of the ecosystem dynamics, where total nitrogen nutrient and phosphate were linked by the Redfield ratio and indifferently represented by one model variable. Da Cunha et al. (2007) used an ocean biogeochemistry model to analyze the impact of river nutrient fluxes (N, Si, Fe and Carbon) on the global and coastal ocean primary production, but concentrated on a short time period of a few decades, likely not long enough to study the feedbacks of the N cycle in the open ocean, considering that the mean residence time of fixed nitrogen in the ocean has been estimated to be a few thousand years (Gruber, 2004).

More recently, Lacroix et al. (2020) implemented estimated riverine nutrient loads in a global ocean model to analyze their implications for global oceanic nutrient concentrations, primary production and $CO_2$ fluxes. Their focus was on pre-industrial nutrient input from rivers, estimated as a function of precipitation, surface runoff and temperature. N was calculated from the simulated P using a fixed N:P ratio, but Lacroix et al. (2020) did not analyze the N cycle feedbacks.

For our study, we used the Earth system climate model of intermediate complexity of the University of Victoria (UVic), Version 2.9 Eby et al. (2009); Weaver et al. (2001). Earth System Models of Intermediate Complexity (EMICs) have been developed to fill the gap between more abstract conceptual models and comprehensive global and Earth system models (ESM) (Claussen et al., 2002). EMICs allow the integration of a large number of processes, more than conceptual models, using often coarser resolution and simplifying assumptions e.g. describing the atmospheric circulation compared to ESMs, and thus substantially reducing computational costs. UVic has been developed as a tool helping to understand processes and feedbacks operating within the climate system on decadal and longer timescales (Weaver et al., 2001).

Atmospheric deposition is known to be another important source of N to the ocean. Although it is estimated to add nitrogen at the same magnitude as the rivers and will also become more important with increasing anthropogenic activities (e.g. Tyrell, 1999; Cornell et al., 1995), it will not be considered in this study. Previously, Landolfi et al. (2017) and Somes et al. (2016) used UVic to study the response of the marine N cycle to idealized atmospheric N deposition and its impact on marine productivity. While Somes et al. (2016) performed a series of idealized sensitivity experiments to evaluate the spatial and temporal scales of N cycle feedbacks, Landolfi et al. (2017) used an atmospheric N deposition forcing reconstructed using the multimodel mean of the Atmospheric Chemistry and Climate Model Intercomparison Project (Lamarque et al., 2013). Both found that N cycle

feedbacks stabilize the model's marine N inventory and limit global changes to the marine N cycle and productivity. But none of these studies included riverine N supply.

In order to disentangle the effects of the different sources of N, we are using UVic without atmospheric N deposition, to focus on the marine biogeochemical response to riverine N inputs to the coastal ocean. To do this we make use of modeled estimates of riverine DIN export from watersheds (Mayorga et al., 2010). While atmospheric deposition is more spread out over the whole ocean, river export of dissolved N reaches the ocean as point sources at different locations and in different concentrations. The global amount of N added to the ocean is comparable between our river-supply study and the atmospheric supply ones by Somes et al. (2016) and Landolfi et al. (2017). Nevertheless, we hypothesize that the response of the marine ecosystem differs with highly concentrated nutrient injections associated with individual rivers.

In order to test the N cycle mechanisms and feedbacks found and described before, we set up an experiment where we simulate differential riverine nitrogen supply to the coastal oceans. Although riverine nitrogen supply is highly influenced by anthropogenic activities, our focus is on the natural nitrogen cycle. Analogous to earlier studies (Lacroix et al., 2020; Da Cunha et al., 2007), we first evaluate the global N inventory and marine primary production after sustained addition of riverine DIN. In a second step, we additionally performed a series of experiments, where we study the responses of the ocean to riverine nutrient supply to individual regions, in order to find out, if N impacts the global ocean differently, depending on the region where river supply takes place.

## 2 Model description and experimental design

Nutrients from the Global Nutrient Export from Water-Sheds (NEWS) 2 model (Mayorga et al., 2010) are added to the University of Victoria Earth System Climate Model (UVic) 2.9 (Keller et al., 2012; Eby et al., 2009; Weaver et al., 2001). The model is outlined, before describing the NEWS2 data set and our experimental design below.

### 2.1 The Earth System Model UVic 2.9

UVic (Weaver et al., 2001) version 2.9 (Keller et al., 2012; Eby et al., 2009) is an Earth System Model of intermediate complexity (Claussen et al., 2002). It consists of a three-dimensional ($1.8°$ x $3.6°$, 19 levels) general circulation model of the ocean, a two-dimensional, single-layer energy-moisture balance atmospheric model, a dynamic-thermodynamic sea ice model, and a terrestrial vegetation model.

The atmospheric component dynamically calculates heat and water fluxes between the atmosphere and the ocean, land and sea ice, and is forced by monthly climatological winds prescribed from NCEP/NCAR. The nineteen vertical levels of the oceanic component, Modular Ocean Model 2 (MOM2), are 50 m thick near the surface and up to 500 m in the deep ocean. The oceanic physical settings are the same as in Keller et al. (2012). The marine ecosystem module of UVic is based on Keller et al. (2012) with updates of some of the equation parameters as noted in Partanen et al. (2016), where a small error in the code was corrected. Seven prognostic variables are embedded within the ocean circulation: two phytoplankton classes (nitrogen fixing diazotrophs

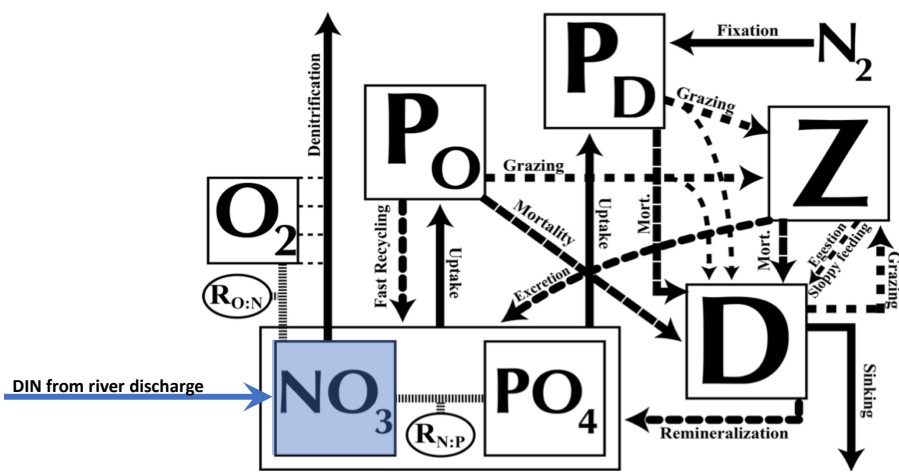

**Figure 1.** Ecosystem model schematics for the NPZD model with the prognostic variables (in square boxes) and the fluxes of material between them, indicated by arrows. The prognostic variables include two nutrients, nitrate ($NO_3$) and phosphate ($PO_4$), two phytoplankton (nitrogen fixers $P_D$ and other phytoplankton $P_O$) as well as zooplankton (Z), sinking detritus (D), and dissolved oxygen ($O_2$). Nitrate ($NO_3$) and phosphate ($PO_4$) are linked through exchanges with the biological variables by constant (Redfield) stoichiometry of organic matter. See further details in the text. Figure updated from Keller et al. (2012).

$P_D$ and other phytoplankton $P_O$), zooplankton (Z), sinking particulate detritus (D), nitrate ($NO_3$), phosphate ($PO_4$) and oxygen
($O_2$) (Fig. 1). $NO_3$ and $PO_4$ are linked through exchanges with the biological variables by constant (Redfield) stoichiometry of organic matter. All biological variables as well as the detritus are expressed in terms of nitrogen (mmol N m$^{-3}$), using Redfield stoichiometry to calculate carbon and P. Since diazotrophs can fix nitrogen gas dissolved in seawater, they are not limited by $NO_3$ nor by a maximum $NO_3$ concentration, while the growth of other phytoplankton is limited by $NO_3$ and $PO_4$ (note that both are additionally limited by iron, light and temperature). The explicit integration of diazotrophs permitting the computation
of nitrogen fixation, is not given in all ocean models but makes UVic a good choice to study nitrogen cycle feedbacks. The maximum potential growth rate of diazotrophs is not only based on temperature as in most models, but also on dissolved iron, which is necessary e.g. for photosynthesis or the reduction of nitrate to ammonium (Keller et al., 2012; Galbraith et al., 2010). Keller et al. (2012) found that the observational estimates were within the range of global nitrate fixation rates from estimations and the patterns of $N_2$ fixation from the new model were mostly consistent with the relatively sparse available observations
(Sohm et al., 2011). See Keller et al. (2012) for a full description and evaluation of simulated marine biogeochemistry.

In the global ocean, fixed N is regulated by the major input fluxes, $N_2$ fixation and riverine input, and the major removal flux, denitrification (here implicitly including anammox). Benthic denitrification, in particular, is believed to be the major sink

for fixed N (Voss et al., 2013; Galloway et al., 2004). It is included here through empirical transfer functions derived from benthic flux measurements (Bohlen et al., 2012). The functions are based on dynamic vertically integrated sediment models and estimate denitrification from the rain rate of particulate organic carbon to the seafloor and bottom water $O_2$ and $NO_3$ concentrations. Like Somes and Oschlies (2015) and Somes et al. (2013) we use a subgrid bathymetry scheme for shallow continental shelves and other topographical features that are too fine to be resolved on the coarse UVic grid, in order to better resolve particulate organic matter sinking and remineralization at the seafloor. For each cell near the coast this scheme calculates the sea floor area within the cell at a higher resolution following Somes et al. (2010b).

## 2.2 Including riverine nutrient supply to the UVic ocean

### 2.2.1 Global Nutrient Export from WaterSheds 2 : NEWS2

Riverine N added to UVic has been generated by a global, spatially explicit model of nutrient exports by rivers. NEWS2 (Mayorga et al., 2010) is the second version of a system of sub-models, which estimate present-day annual export yield for each river basins (kg N $km^{-2}$ $yr^{-1}$) at the river mouths for dissolved and particulate forms of organic and inorganic N and P, as well as dissolved organic and particulate carbon. See Mayorga et al. (2010) for more details on the model configuration. Each sub-model predicts river export of a nutrient element for the base year 2000. This export is calculated as a function of natural and anthropogenic biogeophysical properties of each of the 5761 exoreic basins considered (Seitzinger et al., 2005). NEWS-DIN includes DIN from sewage point sources, as well as N from diffuse sources, mobilized from watershed soils and sediments (Dumont et al., 2005). Despite uncertainties and errors, NEWS-DIN predicts 54-78 % of the variability in DIN export yield (kg N $km^{-2}$ $yr^{-1}$) and 72-83 % of DIN export load (kg N $basin^{-1}$ $yr^{-1}$) of the validation data set used by Dumont et al. (2005). Note that NEWS2 excludes runoff from the Antarctic continent.

### 2.2.2 NEWS-DIN for UVic

To estimate total export per river mouth, we multiplied the yields (kg N $km^{-2}$ $yr^{-1}$) of DIN and dissolved organic nitrogen (DON) by the respective basin area (in $km^2$). Data from the NEWS2 models have been interpolated on the coarser UVic grid and the total exports per river basins have been added to the nearest discharge points, as not every river mouth from NEWS2 has its equivalent discharge point in UVic (Fig. 2). Because there can be strong seasonal variations in nutrient fluxes and fluvial nutrient imports can have different effects on the biogeochemistry of a coastal ecosystem depending on the timing of the fluxes (Eisele and Kerimoglu, 2015; Holmes et al., 2012; Townsend-Small et al., 2011), we used the seasonally cycling climatology of freshwater runoff from UVic to estimate seasonal variations in N supply. Although freshwater discharge and riverine nutrients export are not always correlated, the discharge has an important impact on the nutrient loads of rivers (e.g. Lu et al., 2011, 2009; Sigleo and Frick, 2007; DeMaster and Pope, 1996). Here, we assumed a constant seasonal cycle in runoff and that nitrogen concentrations in the discharged river water are constant throughout the seasonal cycle. We then distributed the annual load over the months, weighted by the fraction of monthly freshwater discharge.

Riverine phosphorus is not added in this experiment and we assume a fixed marine P inventory, like in most previous studies

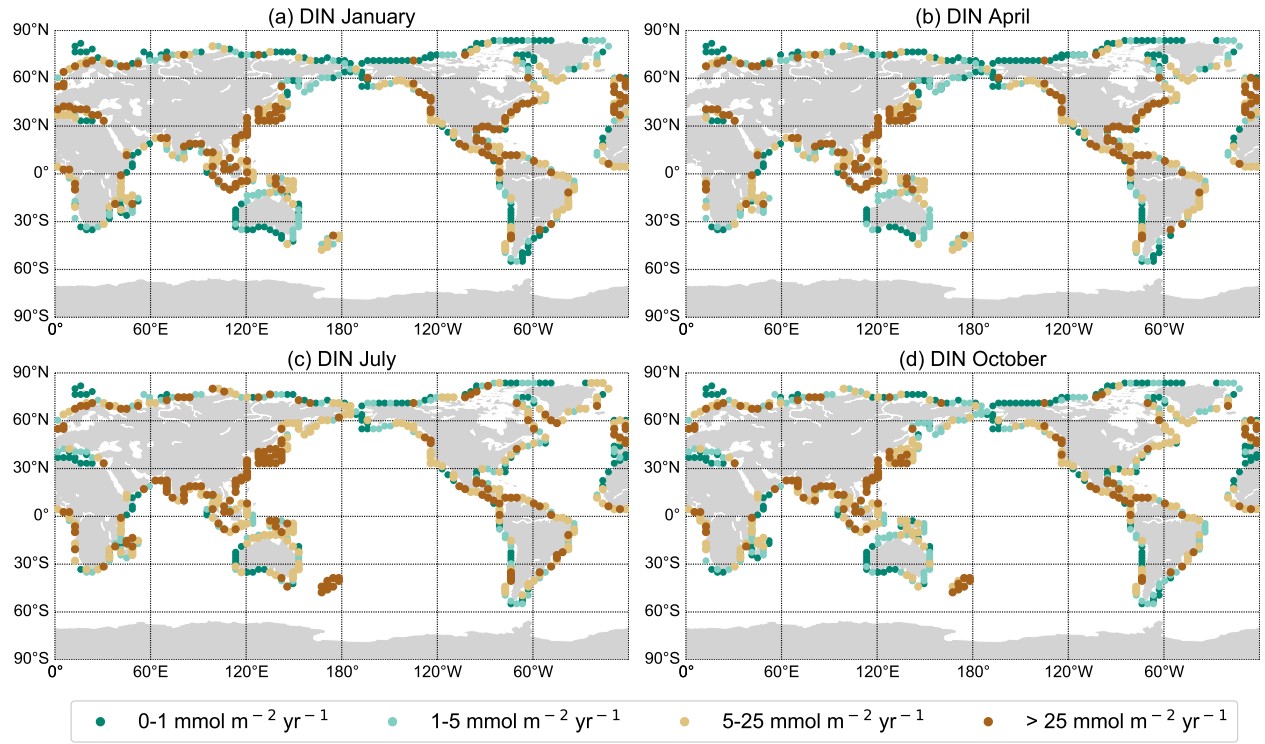

**Figure 2.** DIN export yield for each discharge point in mmol $m^{-2}$ $yr^{-1}$ from NEWS2 data set interpolated on the UVic grid for January (a), April (b), July (c) and October (d).

with UVic. The inclusion of a dynamic P cycle (like in Niemeyer et al., 2017, Kemena et al., 2019) with riverine P supply from NEWS2 will be subject to a follow up study.

### 2.3 Experimental design

To analyze the effect of riverine nutrient export in the UVic model, three experiments have been performed: NEWS, DIN+DON and 2xDIN (Table 1). All simulations were run for 10000 years with benthic denitrification and subgrid bathymetry, starting from an already spun-up steady state with the standard model version (i.e., with no riverine nutrient input) with pre-industrial conditions for insolation and a fixed atmospheric $CO_2$ concentration of 283 ppm (Keller et al., 2012). In NEWS only DIN from NEWS2 was added to the discharge points in UVic. Here, we evaluate how the model and especially the nitrogen cycle react to the riverine nutrient input. In DIN+DON we added DIN and DON from NEWS2. In 2xDIN twice the yield of DIN from NEWS2 has been added. These simulations with increased N supply are scaling experiments to test the mechanisms and feedbacks described for example by Landolfi et al. (2017) and Somes et al. (2016). For comparison, a control simulation has been run for 10000 years without riverine DIN supply (referred to as CTR). Globally, NEWS2 predicts a riverine N supply of 22.8 Tg N $yr^{-1}$ for DIN and 11.8 Tg N $yr^{-1}$ for DON. Both enter the biogeochemical model as $NO_3$ fluxes in mol N $m^{-2}$ $s^{-1}$,

**Table 1.** UVic simulations and global annual nutrient flux from river discharge from Global Nutrient Export from WaterSheds 2 (Mayorga et al., 2010)

| Simulation | Global N-flux [Tg N yr$^{-1}$] | Short description |
|---|---|---|
| CTR | 0.0 | Control simulation without riverine nutrient supply |
| NEWS | 22.8 | Riverine DIN input from NEWS2 |
| DIN+DON | 34.6 | Riverine DIN + DON input from NEWS2 |
| 2xDIN | 45.6 | Twice the riverine DIN input from NEWS2 |

thereby implicitly assuming that all DON is bioavailable or rapidly turned over to DIN. The marine ecosystem dynamics as well as the biogeochemical cycles of the model run have been evaluated in previous studies under the standard boundary conditions, without riverine nutrients (e.g. Somes et al., 2013; Keller et al., 2012; Somes et al., 2010b; Schmittner et al., 2008, 2005). We therefore concentrate on the evaluation of the response of the marine biogeochemical model to the new model component of riverine nutrient discharge. The global ocean biology is reacting to the new N cycle components in the first three to four thousands years of the simulations. After this first phase, the N budget is slowly equilibrating towards a steady state (see Figure in supplements). For the evaluation of the resulting ocean biogeochemistry, we analyze in the following the mean of the last 100 years of each 10 000 years long simulation.

## 3 Results and discussions

### 3.1 Nitrogen

#### 3.1.1 Global nitrate distribution

In comparison with observational data of the World Ocean Atlas (Garcia et al., 2019), the model simulates the general structure of the profiles fairly well, but underestimates the observed $NO_3$ in the water column in each ocean basin by 3 to 4 mmol m$^{-3}$, especially omitting the midwater maximum around 1000 m (Fig. 3). Global average $NO_3$ concentrations only vary a little between the simulations (from 22.19 mmol m$^{-3}$ in CTR to 22.48 mmol m$^{-3}$ in NEWS and 22.84 mmol m$^{-3}$ in 2xDIN) and the differences to CTR correspond to +1.1 %, +1.8 % and +2.5 % of the total observed inventory for NEWS, DIN+DON and 2xDIN, respectively. Nevertheless, in all three simulations (NEWS, DIN+DON, and 2xDIN), $NO_3$ concentrations are globally higher compared to CTR (Fig. 5, Fig. 6). The absolute error between model and observations decreases with higher riverine N supply (top right panel in Fig. 3). At the surface, the global ocean $NO_3$ distribution patterns are very similar between the model and the observations, as well as between the control (CTR) and the NEWS simulation (Fig. 4).

The supply of riverine $NO_3$ affects the ocean nutrient concentration not only locally near the river mouths, but also in regions far away from the coasts. Surface $NO_3$ concentrations increase with higher river supply in the coastal regions and in the higher

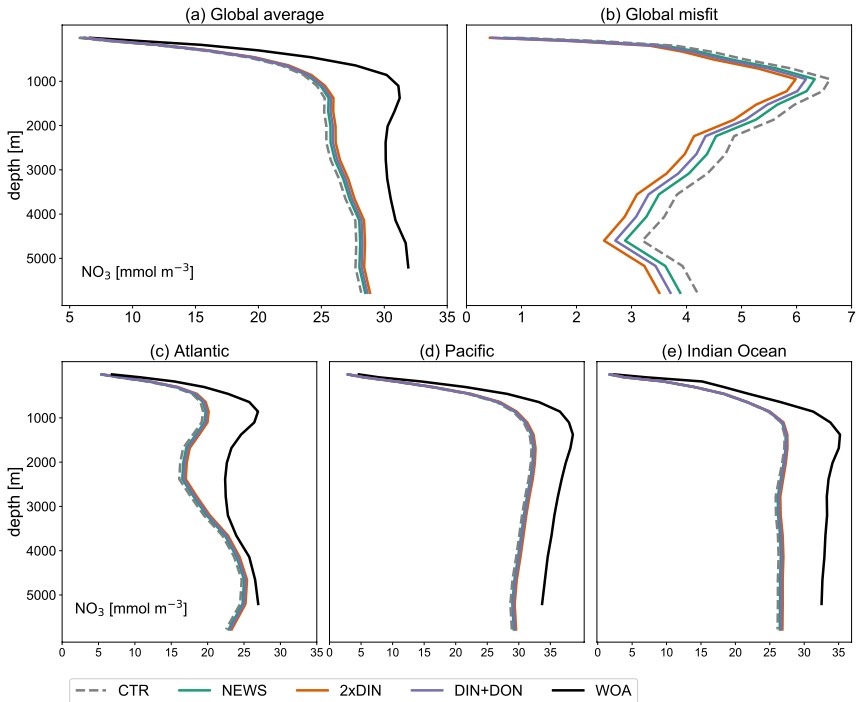

**Figure 3.** Profiles of global averaged $NO_3$ in mmol m$^{-3}$ from UVic simulations with and without riverine DIN export from NEWS data sets and observations with the WOA. (a) Global ocean average of $NO_3$. (b) Global profiles of misfit in $NO_3$ compared to the observations. (c)-(e). $NO_3$ profiles for the three main ocean basins, Atlantic (ATL), Pacific (PAC) and Indian Ocean (IND).

latitudes. The globally highest increase in $NO_3$ can be found in the 2xDIN experiment (see also table 2) and the $NO_3$ increase is higher in the deeper ocean than at the surface. Interestingly, the increase in the oceanic N inventory is more than twice as high in UVic 2xDIN compared to UVic NEWS, indicating non-linear feedbacks.

While higher $NO_3$ concentrations due to riverine input are not entirely surprising, some regions present however lower concentrations compared to CTR. In all simulations $NO_3$ is slightly lower at the surface in large parts of the tropical and subtropical

oceans. At 850 m depth, the ocean loses $NO_3$ upon the addition of riverine N in our simulations in low oxygen regions where denitrification occurs, such as the Gulf of Benguela, the Bay of Bengal and the eastern equatorial Pacific near the coast of Central America (Fig. 6).

In major parts of the Atlantic and Pacific Ocean basins, $NO_3$ concentrations are higher in NEWS than in CTR (Fig. 5). $NO_3$ is particularly elevated in the upper 2000 m in the North Atlantic Ocean (up to 2 mmol N m$^{-3}$ in 2xDIN, corresponding to +18

215  %) and upper 1000 m in the North Pacific, but the difference between the simulations is positive in the whole basins, indicating that a substantial part of the additional riverine N is exported into the open and deeper ocean. At the surface of the tropical and subtropical oceans, however, $NO_3$ concentrations are lower by maximal 0.9 mmol N m$^{-3}$ in the UVic-NEWS experiments compared to CTR (Fig. 6).

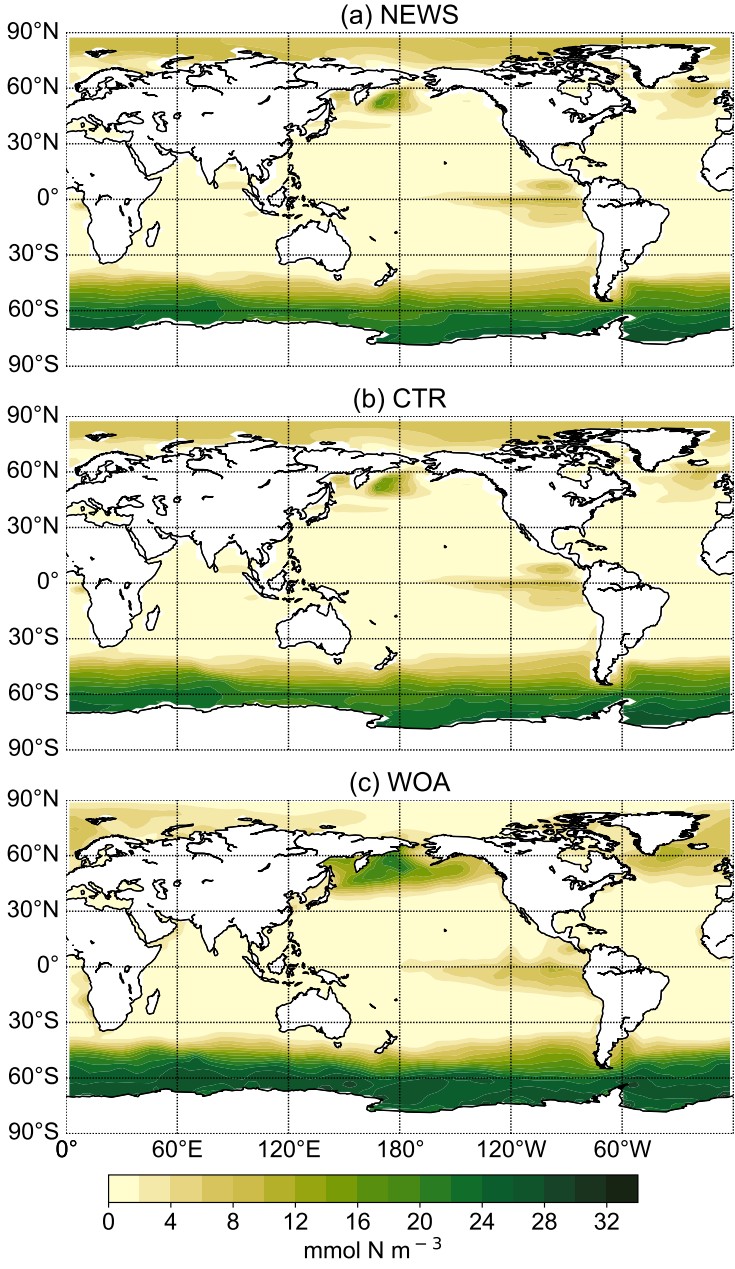

**Figure 4.** Surface NO$_3$ concentrations for simulations UVic NEWS (a), UVic control (b) and from observations from World Ocean Atlas (Garcia et al., 2019) (c).

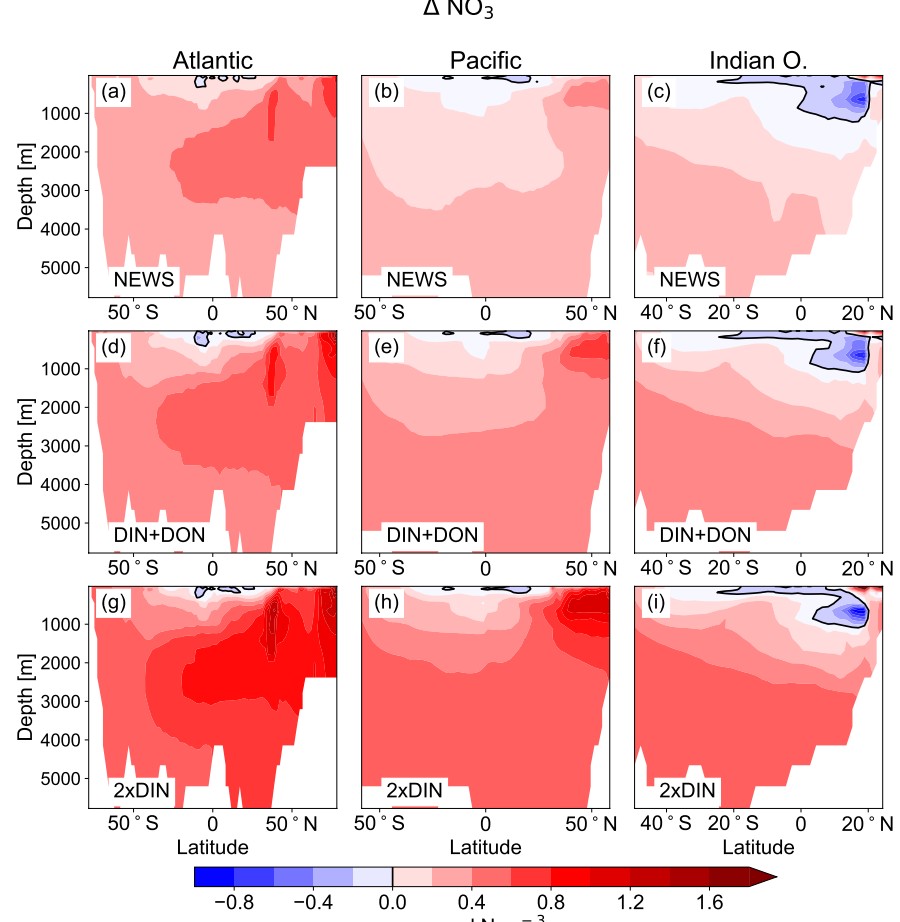

**Figure 5.** Difference in zonal mean ocean concentrations of $NO_3$ between the UVic simulations with riverine DIN export and the control simulation. a,b,c : difference between NEWS and CTR; d,e,f : difference between DIN+DON and CTR; g,h,i: difference between 2xDIN and CTR. The columns show the zonal mean of the Atlantic (a,d,g), the Pacific (b,e,h) and the Indian Ocean basins (c,f,i). The difference in zonal averaged $NO_3$ concentrations are higher than the colorbar maximum at the surface in the northern Indian Ocean basin with a maximum for 2xDIN at 7.2 mmol N $m^{-3}$. Note that the three ocean basins have different sizes in terms of latitudes but for layout reasons the panels have the same dimensions.

The Indian Ocean basin comprises the Arabian Sea and the Gulf of Bengal. Zonally averaged $NO_3$ concentrations reflect essentially the behavior of the Bay of Bengal, where the rivers of the Ganges Delta supply high amounts of nutrients to the northern basin (Fig. 5, Fig. 7). Here, the model simulates high $NO_3$ concentrations at the surface (in the north, several hundred percent higher in NEWS when compared to CTR). But in the deeper northern Indian Ocean basin down to approximately 2000 m, $NO_3$ concentrations are significantly lower in NEWS than in CTR. Considering the zonal average of the Indian Ocean, $NO_3$ concentrations are lower by 0.7 to 0.9 mmol N $m^{-3}$, and even more if only the zonal average of the Bay of Bengal is

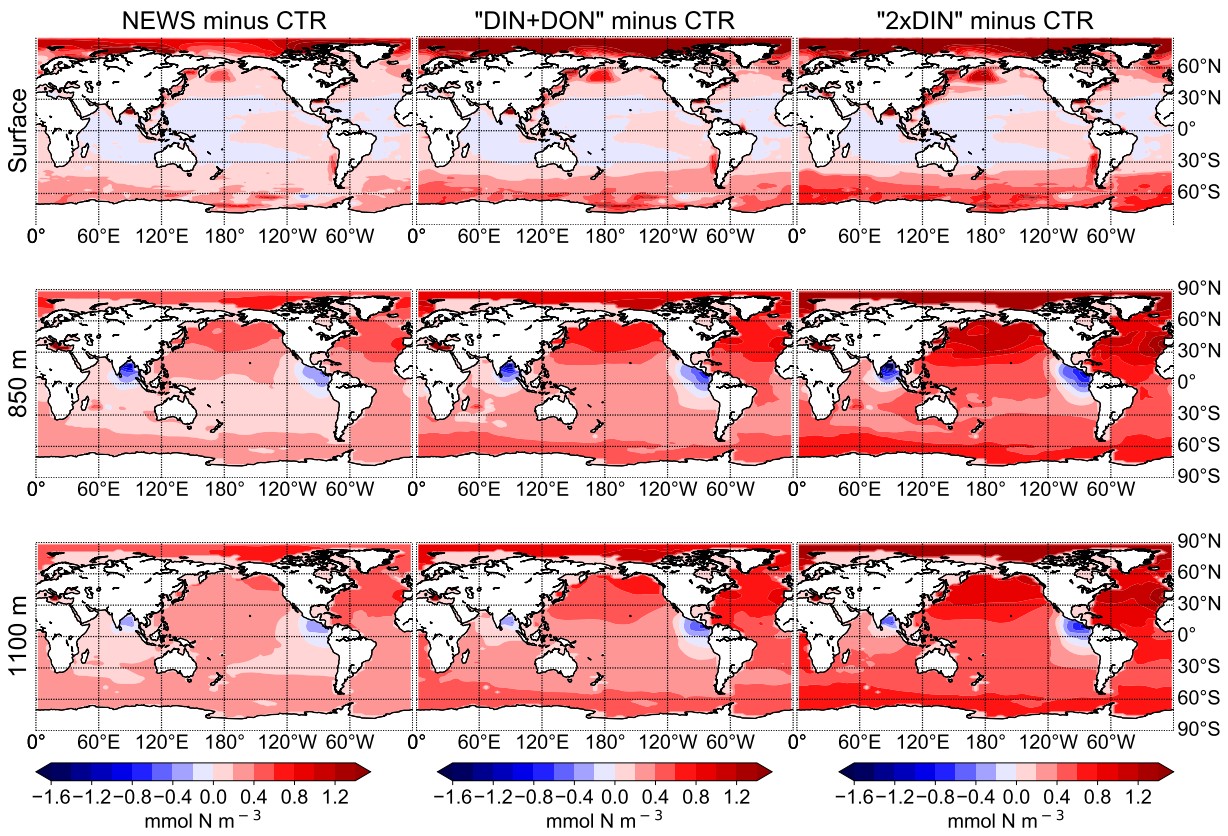

**Figure 6.** Difference in global distribution of NO$_3$ concentrations between experimental simulation and the control simulation in mmol N m$^{-3}$; The different simulations are NEWS, DIN+DON and 2xDIN. The differences are shown at three different depths: Surface (top), 850 m (middle) and 1100 m (bottom).

considered.

Part of the global NO$_3$ patterns can be explained by the interaction of ocean circulation and biology. N is transported into the interior ocean via circulation and also accumulates due to the biological pump. But these processes do not explain the loss in N in the subtropical surface oceans or the Bay of Bengal. The same applies to the total amounts of N. Despite the continuous supply from the rivers, the additional NO$_3$ in NEWS, DIN+DON and 2xDIN compared to CTR amounts to an increase of only

to 1.1 %, 1.8 % and 2.5 %, respectively (Table 2). What limits the increase in the global oceanic N inventory is the combination of the N cycle processes denitrification and N$_2$ fixation.

**Table 2.** Amount of additional nitrogen in Tg N at the end of each simulation compared to CTR

| Simulation | Glob. NO$_3$ concentr. | Total N added | Change in N inventory | Change in N inventory relative to CTR | change in N inventory relative to total N addition |
|---|---|---|---|---|---|
| | [mmol N m$^{-3}$] | [Pg N] | [Tg N] | [%] | [%] |
| CTR | 22.19 | 0.00 | 0.00 | 0.00 | 0.00 |
| NEWS | 22.48 | 228 | 5278 | +1.12 | +2.31 |
| DIN+DON | 22.66 | 346 | 8298 | +1.77 | +2.40 |
| 2xDIN | 22.84 | 456 | 11895 | +2.53 | +2.61 |
| WOA | 26.29 | - | - | - | - |

### 3.1.2 Denitrification and nitrogen fixation

Denitrification is known to be the main sink for fixed N in the ocean (Gruber, 2004; Codispoti et al., 2001). It occurs both in marine sediments and in the water columns under suboxic conditions, for example in the simulated Bay of Bengal. As a result of these dynamics, if N is added via river discharge, UVic simulates globally higher water column and benthic denitrification rates (Table 3). Note that in these simulations global benthic denitrification and global water column denitrification amount to similar magnitudes, indicating the importance of both in the global N-cycle (Somes et al., 2016, 2013). For both processes estimates vary considerable: for water column denitrification estimates are between 50 and 150 Tg N yr$^{-1}$, for benthic denitrification between 100 and 300 Tg N yr$^{-1}$ (Galloway et al., 2004; Gruber, 2004; Bohlen et al., 2012; Somes et al., 2013). In all our simulations, denitrification rates stay in the range assumed for a balanced fixed-N budget in the preindustrial ocean (e.g. Somes et al., 2013). However, benthic denitrification is more evenly distributed than water column denitrification. In order to study regional effects it is helpful to include both processes. Nevertheless, models can have a balanced nitrogen budget without including benthic denitrification.

While the global pattern of denitrification is very similar in the simulations with additional riverine N compared to CTR, in proximity to river discharge points, total denitrification rates are higher by up to 1 mol N m$^{-2}$ yr$^{-1}$ (Figure 9). Somewhat off the coasts however, total denitrification appears lower in the simulations with riverine nutrient supply (by up to 50 mmol N m$^{-2}$ yr$^{-1}$). At the same time, total global N$_2$ fixation rates decrease in all three simulations compared to CTR (Table 3). Nitrogen fixation is a significant process in the marine nitrogen cycle and a major source of nitrogen in the open ocean. Nitrogen fixing organisms are able to convert dissolved nitrogen gas (N$_2$) into ammonia, but are limited in their growth by phosphate and iron (Deutsch et al., 2007; Moore and Doney, 2007; Karl et al., 1997; Redfield et al., 1963), and phosphate is not altered by the additional N from river supply. The global rate and geographical distribution of nitrogen fixation are still uncertain. Observations remain sparse and highly variable in space and time. Combined with insufficient understanding of the controls of marine N$_2$ fixation, this results in high uncertainties in the global pattern of marine nitrogen fixation (Wang et al., 2019; Landolfi et al., 2018; Somes et al., 2013). Deutsch et al. (2007) and Luo et al. (2012) estimated a global nitrogen fixation rate

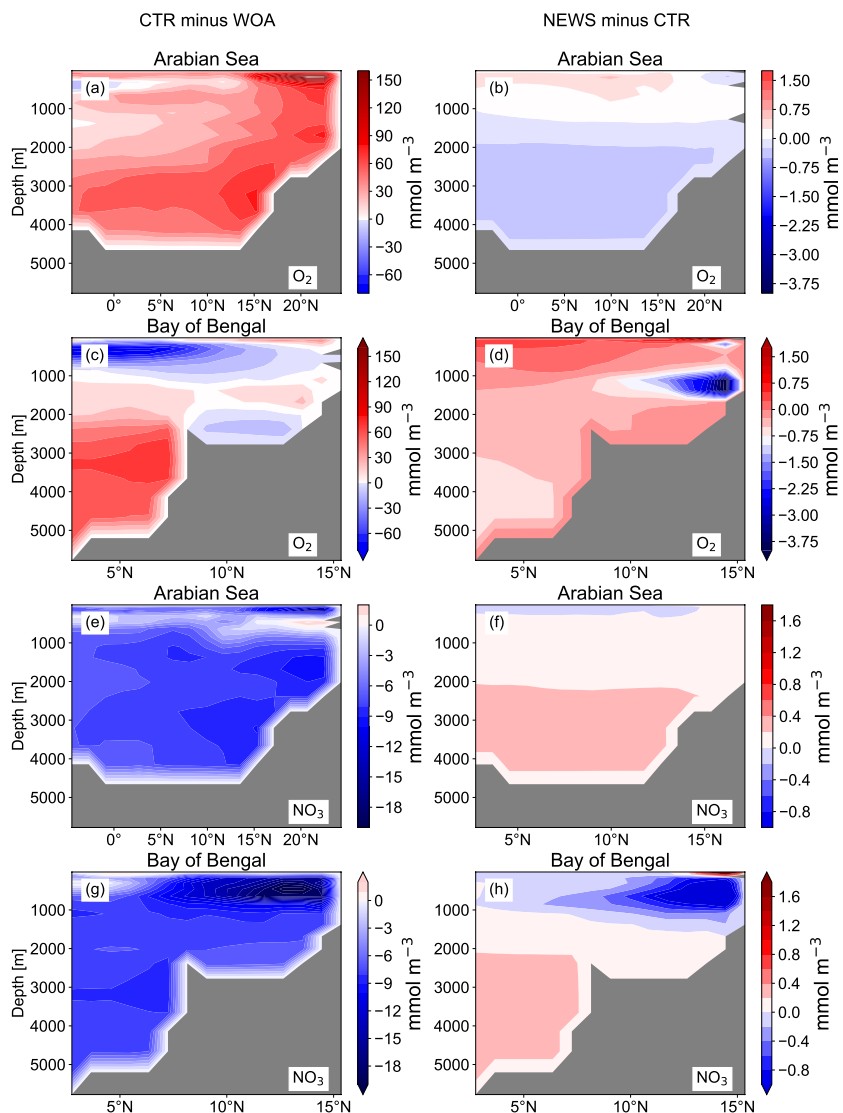

**Figure 7.** Difference of zonal average of $NO_3$ and $O_2$ in the basins of the Arabian Sea (a, b, e, f) and the Gulf of Bengal (c, d, g, h). The panels in the left column show the difference between CTR and World Ocean Atlas (WOA), in the right column the difference between NEWS and CTR. All results are shown in mmol m$^{-3}$.

of 140 Tg N yr$^{-1}$ and most recent studies stay in this range, although some studies suggest, that the global rates could be much

higher (Wang et al., 2019; Landolfi et al., 2018; Somes et al., 2013; Karl et al., 2002).

The global rates calculated from our experiments with UVic (Table 3) are also higher than the estimates from Deutsch et al. (2007) and Luo et al. (2012). Although previous studies with UVic have given rates of $N_2$ fixation between 128 and 150 Tg N yr$^{-1}$ (Landolfi et al., 2017; Keller et al., 2012), the CTR simulation in our configuration estimates global $N_2$ fixation rates of 219 Tg N yr$^{-1}$. In our case, this is due to the additional integration of benthic denitrification, which has not always been considered in previous UVic studies. The additional N sink in form of benthic denitrification promotes conditions that favor N-fixers, i.e., diazotrophs, leading to higher nitrogen fixation rates.

In the UVic CTR simulation, $N_2$ fixation is mostly confined to the tropical and subtropical oceans and is especially concentrated in the northern Indian Ocean, the eastern Pacific and the eastern Atlantic Ocean (Fig. 8a). This is comparable to the distribution in Keller et al. (2012) and Somes et al. (2010a), both using UVic in different configurations. The patterns of $N_2$ fixation are therefore consistent with observations, as far as they are known, with the same limitations as for Keller at al. (2012) and Somes et al. (2010a). For example, in the subtropical North Atlantic, where some of the highest rates of $N_2$ fixation have been measured (Capone et al., 2005), UVic simulates almost no $N_2$ fixation at all. The simulation NEWS, DIN+DON and 2xDIN show, that adding riverine N leads to a net decrease in $N_2$ fixation in nearly the whole area, where it occurs, but especially near the river mouths (Fig. 8b-d). The main regions, where $N_2$ fixation is significantly decreasing are the Gulf of Guinea, the Gulf of Bengal and near the Amazon River mouth.

In a previous study with UVic, Somes et al. (2016) have shown that increasing atmospheric N deposition could lead to a reduction in $N_2$ fixation, due to non-nitrogen-fixing phytoplankton being more competitive than N fixers, when key nutrients like iron and phosphate are limiting. Here, it is the input of riverine nitrogen that stimulates the reduction in $N_2$ fixation locally, where N reaches the ocean. Reductions in $N_2$ fixation can then partly explain the lower $NO_3$ concentrations at the surface of the tropical and subtropical oceans in NEWS, even though these areas are far from riverine N input (see 3.1.1. and Figure 6, first row). Note that these results show the distribution at steady state after 10000 years of riverine nitrogen supply. Not all fixed nitrogen is consumed by biological activity, but part of the additional N is also transported with ocean circulation and can "replace" N from nitrogen fixation in regions far off the coast, leading to decreasing $N_2$ fixation at the surface of the tropical and subtropical oceans.

### 3.1.3 The N-cycle feedback mechanisms

The interaction between $N_2$ fixation, denitrification, and riverine nitrogen supply, can also explain the significant loss in $NO_3$ in some regions localized before: the Gulf of Guinea, the Gulf of Bengal and the western coast of Central America (Fig. 5 and Fig. 7). In addition, these three regions have also in common that they are known to have very low oxygen concentrations. Note, that the global volume of ocean minimum zones, defined here as regions with $O_2$ concentration lower than 70 $\mu$mol kg$^{-1}$, is increasing with higher nitrogen supply, from $52*10^6$ km$^3$ in CTR, to $54*10^6$ km$^3$ in NEWS, $56*10^6$ km$^3$ in DIN+DON and $58*10^6$ km$^3$ in 2xDIN. In the Bay of Bengal, oxygen concentrations even though higher at the surface in NEWS than in CTR, are very low in the NEWS simulations in the subsurface waters and the whole deeper basin (Fig. 7). These suboxic waters are

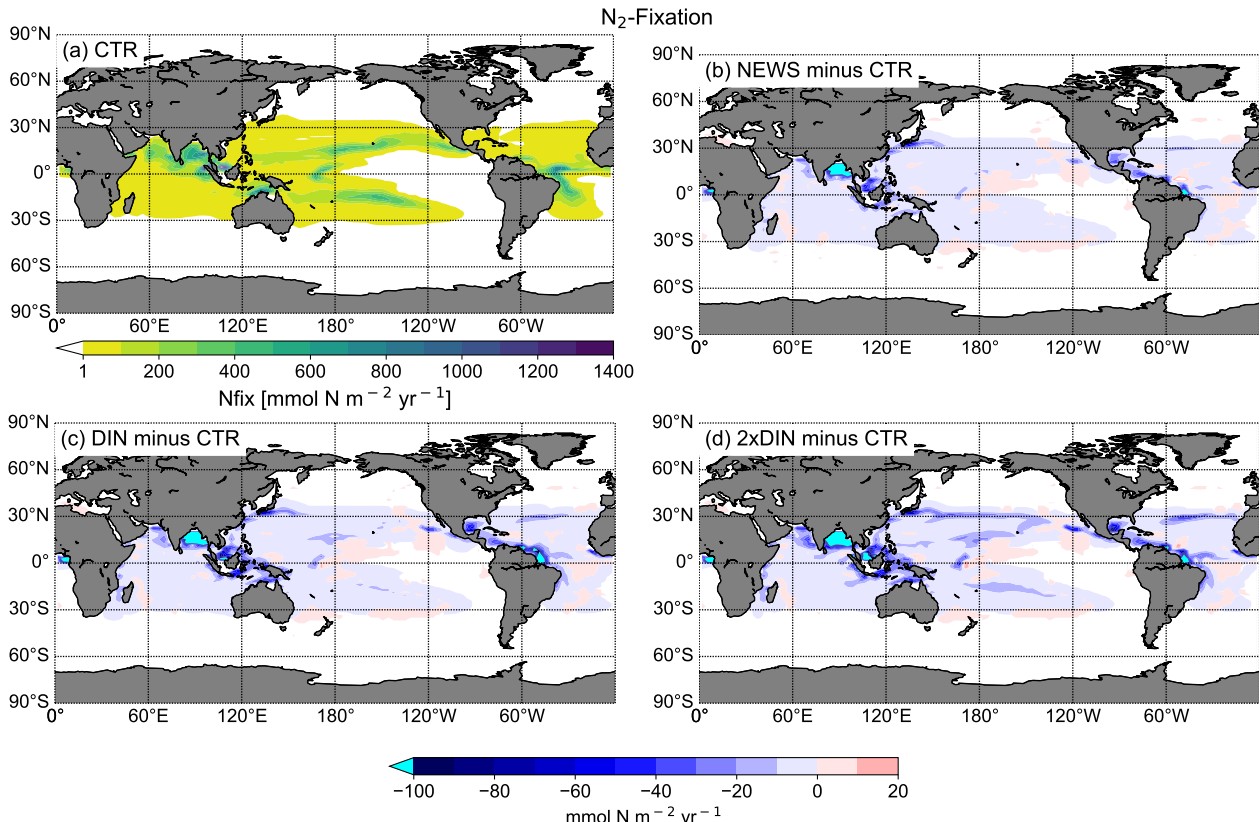

**Figure 8.** (a) Global distribution of $N_2$ fixation in CTR in mmol N m$^{-2}$ yr$^{-1}$. The white areas show regions where rates are smaller than 1 mmol N m$^{-2}$ yr$^{-1}$. (b) Difference in annual vertically integrated rates of $N_2$ fixation calculated from UVic NEWS and CTR in mmol N m$^{-2}$ yr$^{-1}$. (c) Difference in $N_2$ fixation in the simulation with riverine DIN and DON. (d) Difference in $N_2$ fixation in the simulation with twice the amount of DIN. The white areas show regions where differences are smaller than 0.01 mmol N m$^{-2}$ yr$^{-1}$. Local minimas can be found near the Amazon river basin (from -356 mmol N m$^{-2}$ yr$^{-1}$ in NEWS-CTR to -382 mmol N m$^{-2}$ yr$^{-1}$ in DIN-CTR), in the Bay of Bengal (from -347 mmol N m$^{-2}$ yr$^{-1}$ to -646 mmol N m$^{-2}$ yr$^{-1}$ in 2xDIN-CTR) and in the Gulf of Guinea (from -180 mmol N m$^{-2}$ yr$^{-1}$ in NEWS-CTR to -303 mmol N m$^{-2}$ yr$^{-1}$ in 2xDIN-CTR).

furthermore located in proximity to riverine N input and high denitrification rates (Fig. 10). While total denitrification rates (benthic and water column denitrification) are already quite high in CTR, they are further increased in NEWS, DIN+DON and 2xDIN in the northern Bay of Bengal, adjacent to the river delta.

Landolfi et al. (2013) found that the negative feedback mechanism between $N_2$ fixation and denitrification, generally stabilizing the marine N inventory, can turn into a destabilizing positive feedback, generating runaway N loss, if a close spatial association of $N_2$ fixation and denitrification occurs. This is due to the stoichiometric imbalance created by the combination of these processes. Denitrification occurs in anoxic or suboxic environments, where nitrate or nitrite can be used as a substitute

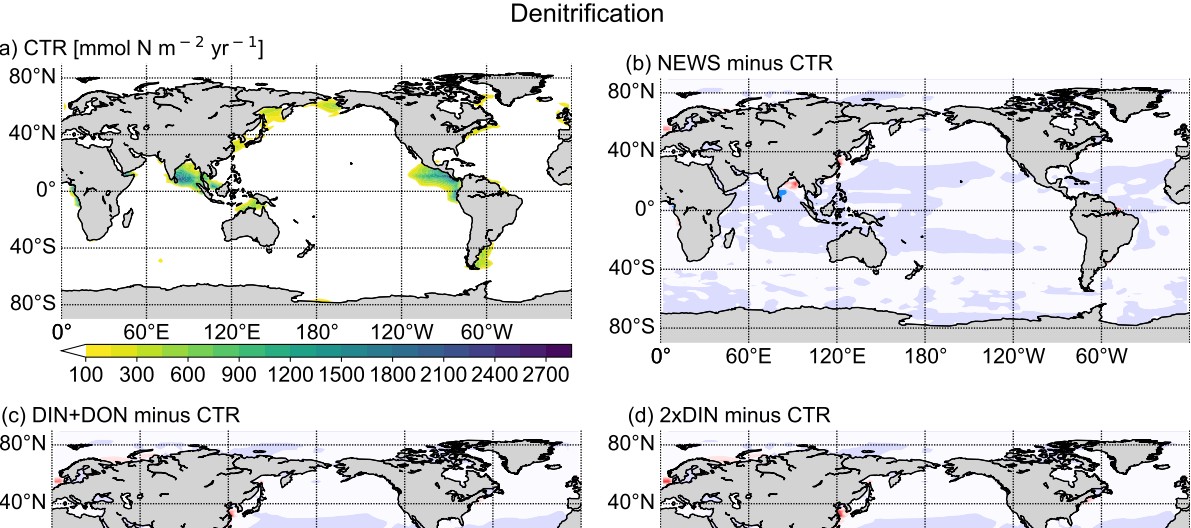

**Figure 9.** (a) Global distribution of total denitrification in CTR in mmol N m$^{-2}$ yr$^{-1}$. The white areas show regions where rates are smaller than 100 mmol N m$^{-2}$ yr$^{-1}$. (b) Difference in annual vertically integrated rates of total denitrification calculated from UVic NEWS and CTR in mmol N m$^{-2}$ yr$^{-1}$. (c) Difference in total denitrification in the simulation with riverine DIN and DON. (d) Difference in total denitrification in the simulation with twice the amount of DIN.

terminal electron acceptor instead of oxygen. Denitrification consumes 7 mol of NO$_3$ for every mole of organic N provided by N$_2$-fixation, and remineralized anaerobically via denitrification. If more than 1/7 of the organic N provided via N$_2$ fixation is denitrified, this leads to a net loss of N by more N lost during denitrification than added via N$_2$ fixation, called 'vicious cycle'
by Landolfi et al. (2013).

In the Bay of Bengal, oxygen concentrations appear higher at the surface in NEWS than in CTR by around 1.5 mmol m$^{-3}$ at least in the southern part of the bay, which could be due to enhanced production. However, compared to WOA, the oxygen concentrations are still very low in the upper 800 m in NEWS like in CTR, and they are particularly low in the NEWS simulations in the subsurface waters and the whole deeper basin (Fig. 7 ). The 'vicious cycle' is triggered here by the input of new
N from riverine export near oxygen minimum zones, explaining the NO$_3$ deficit found in the simulated Bay of Bengal (Fig. 5). Note that UVic, similar to most other biogoechemical ocean models, misplaces the main oxygen minimum zone from the

**Table 3.** Global nitrogen sources (river supply, N$_2$ fixation) and sinks (denitrification) averaged over the last 100 years of the simulations. All fluxes are given in Tg N yr$^{-1}$. Note that the global sums from sources and sinks do not exactly add to zero due to natural variability in the modeled N-cycle.

| Simulation | River supply | N$_2$ fixation | Water column denitrification | Benthic denitrification |
|---|---|---|---|---|
| CTR | 0.0 | 219.1 | 110.9 | 108.7 |
| NEWS | 22.8 | 205.9 | 113.2 | 115.9 |
| DIN+DON | 34.6 | 199.3 | 114.7 | 119.7 |
| 2xDIN | 45.6 | 192.7 | 116.1 | 122.7 |

Arabian Sea to the Gulf of Bengal (Séférian et al., 2020). In reality, high water column denitrification has been observed in the Arabian Sea, while in the Gulf of Bengal highly variable oxygen concentrations seem to inhibit denitrification (Johnson et al., 2019; Bange et al., 2005).

At the end of the simulation, the global marine N inventory is higher by 5278 Tg N in NEWS compared to CTR, which corresponds to 1.12 % of the global N inventory in CTR and 2.3 % of the total riverine N input over the 10000 years of the simulation. Even for the highest scenario (2xDIN), the total increase in global N represents only +2.53 % of the reference N inventory. Most of the additional N input through river discharge is thus compensated for by the feedbacks of the N cycle. However, relative to the total additional input, the N increase in 2xDIN is higher than in NEWS (+2.6 % compared to +2.3 %),

which means that the negative feedbacks do not compensate in 2xDIN as much as in NEWS. A possible reason for this result could be, that the main positive N-loss feedback becomes smaller or that the positive feedbacks, resulting in loss of N, take place in very localized low-oxygen areas, that can not expand further (e.g. Bay of Bengal, Amazon river mouth), while riverine N is supplied through river mouths scattered over the world.

**3.2   Marine primary production**

The rates of simulated annual global net primary production (NPP) compare well to present day estimates of annual global NPP (43.5 to 67 Pg C yr$^{-1}$) derived from satellite measurements (Buitenhuis et al., 2013; Westberry et al., 2008; Carr et al., 2006; Behrenfeld et al., 2005) and vary little between the simulations (Table 4). That is NPP increases only slightly with riverine N supply.

Annually averaged and vertically integrated primary production rates from CTR shows high rates in the equatorial eastern Pacific, Atlantic and Indian Ocean as well as in the upwelling region of the western south Atlantic Ocean (Fig. 12, a.). This global pattern persists in the simulations with riverine N supply (differences are small in Fig. 12, b.-d.). Nevertheless, with rivers supplying N to the ocean, differences are visible at coastal scales: NPP increases locally, close to the river mouths,

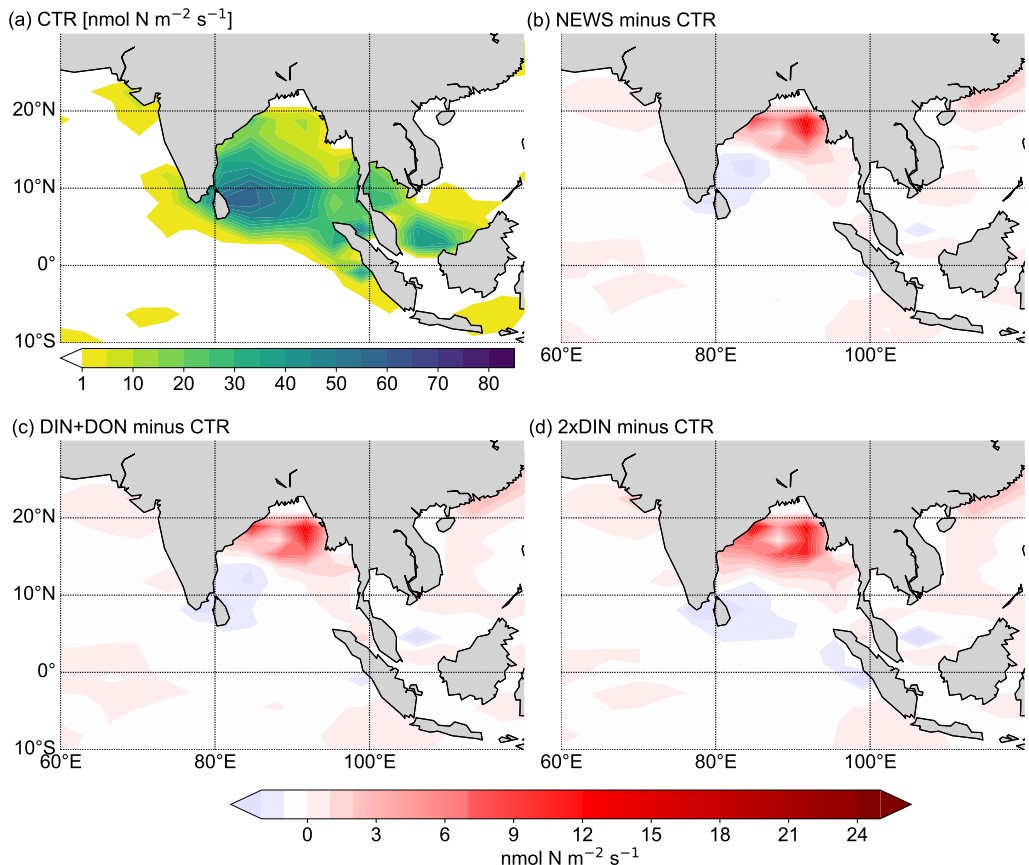

**Figure 10.** Total (water column and benthic) denitrification in the Bay of Bengal (a) Total denitrification from CTR in nmol N m$^{-2}$ s$^{-1}$. In the white areas total denitrification rates are smaller than 1 nmol N m$^{-2}$ s$^{-1}$. (b) Difference in denitrification between NEWS and CTR. (c) Difference in denitrification between DIN+DON and CTR. (d) Difference in denitrification between 2xDIN and CTR.

especially near the coasts of Europe, China, parts of North and parts of South America. Most changes range between -60 and
+100 g C m$^{-2}$ yr$^{-1}$. In isolated regions, marine primary production rates are lower in the three NEWS simulations than in CTR. This is particularly striking in parts of the Bay of Bengal, but also in the equatorial west Atlantic Ocean, to the north of the Amazon river mouth. In large regions of the subtropical and tropical oceans where surface NO$_3$ concentrations in our simulations are lower than in CTR, NPP differences also show a decrease.

In NEWS, where only DIN is supplied by the rivers, NPP increases mainly in shelf and near coastal oceans. The main differences are in the magnitude rather than the patterns of NPP, however, higher amounts of N added by the rivers in DIN+DON and 2xDIN also impact marine productivity in the open ocean, far away from the river mouths. In the western subtropical and tropical waters NPP decreases with higher N input. These regions correspond to the regions where diazotrophs can be found

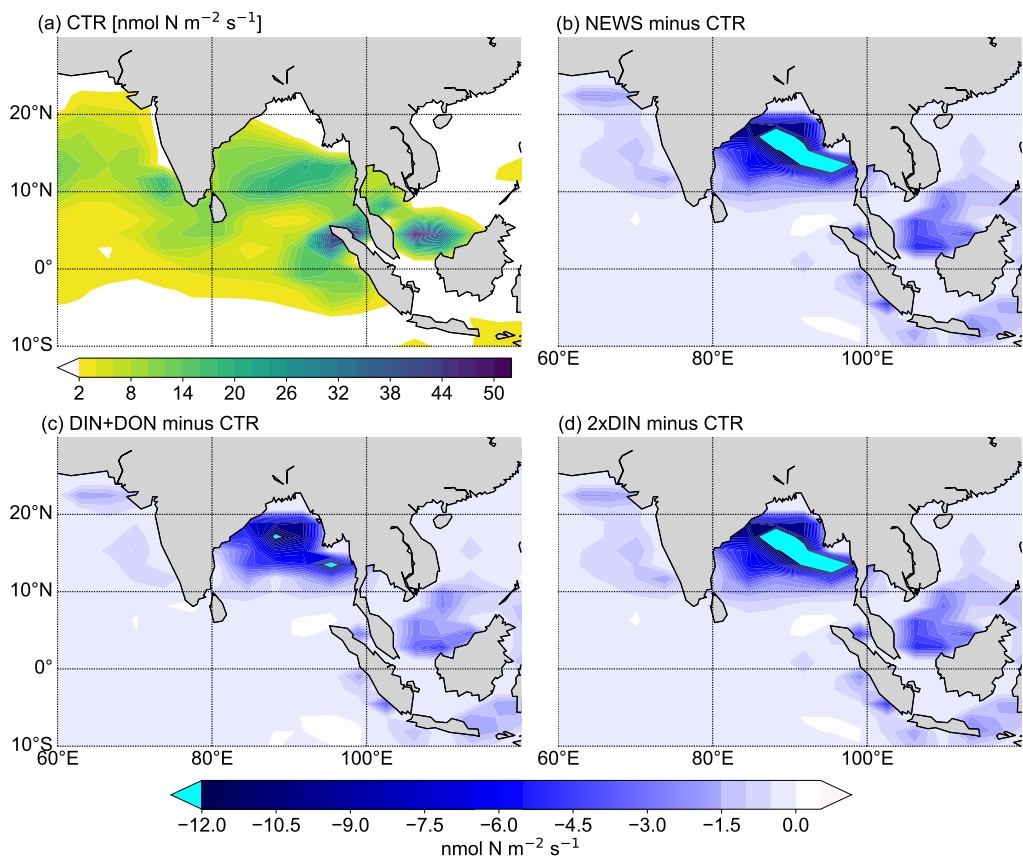

**Figure 11.** $N_2$ fixation in the Bay of Bengal (a) $N_2$ fixation from CTR in nmol N m$^{-2}$ s$^{-1}$. The white color indicates regions where $N_2$ fixation is smaller than 2 nmol N m$^{-2}$ s$^{-1}$. (b) Difference in $N_2$ fixation between NEWS and CTR. (c) Difference in $N_2$ fixation between DIN+DON and CTR. (d) Difference in $N_2$ fixation between 2xDIN and CTR

and where $N_2$-fixation also decreases in our simulations. In the higher northern latitudes primary production is enhanced off the coastal oceans, in the North Atlantic and the western North Pacific Oceans as well as on the Arctic Ocean shelf (Fig. 12d.). Two

"physical" explanations suggest themselves: first, near the coast the riverine N is consumed until phosphate becomes limiting. Then, the excess N is exported from the coastal oceans, leading to higher productivity farther away. Second, decreasing NPP in the open ocean can be the consequence of a seesaw effect, also called "nutrient robbing". Because higher N concentrations increase NPP in the coastal ocean, other nutrients, like P, are also consumed here instead of being exported to the open ocean. This can lead to lower rates of primary production farther away (Giraud et al., 2008).

Globally the differences in NPP in NEWS compared to the control simulation are close to the spatial variance of annually averaged NPP in the model ($\sim$57 g C m$^{-2}$ yr$^{-1}$). Most changes are smaller than $\pm$ 40 g C m$^{-2}$ yr$^{-1}$ ($\pm$ 2 %), even though

NPP

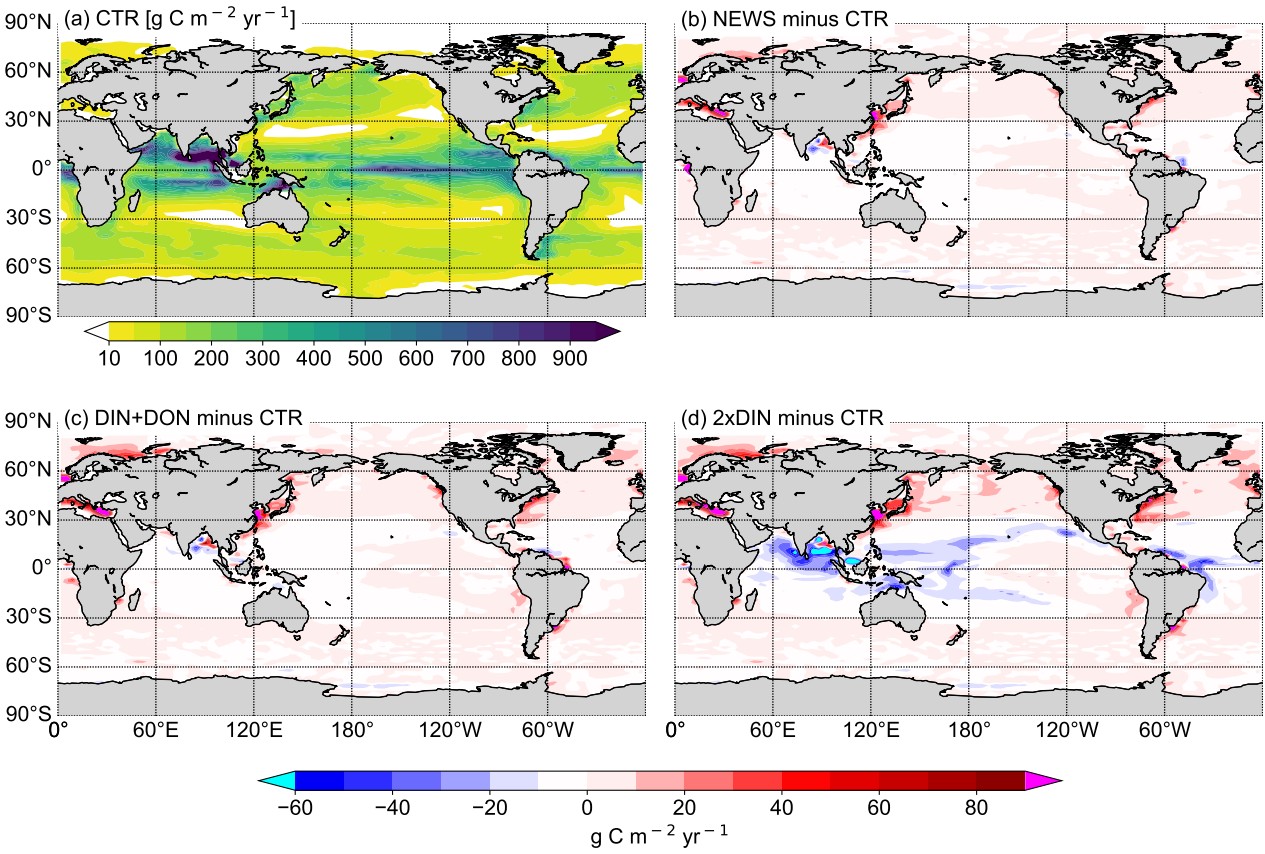

**Figure 12.** (a) Annual vertically integrated rates of primary production (NPP) calculated from UVic CTR in g C m$^{-2}$ yr$^{-1}$. (b)-(d) Differences in NPP distribution calculated from different UVic simulations with riverine N export and from UVic simulation without riverine nutrient input (CTR) in g C m$^{-2}$ yr$^{-1}$. (b) NPP from NEWS with with riverine DIN. (c) NPP in the simulation with riverine DIN and DON. (d) NPP in the simulation with twice the amount of DIN. The extreme values shown in cyan and red in these panels are listed in Table 5.

locally the changes can be high (down to -122 g C m$^{-2}$ yr$^{-1}$ in the Bay of Bengal, or even higher than +500 g C m$^{-2}$ yr$^{-1}$ in the East China Sea). The total increase in NPP varies between +0.7 % (NEWS) and +1.3 % (2xDIN) compared to CTR. These changes reflect in wide parts changes in NO$_3$ due to the riverine inputs, except for the higher latitudes and other regions, where
light-, temperature- or iron-limitation occur. In the higher DIN experiment, NPP is globally a little higher than in the other simulations including CTR, but the distribution shows NPP hot-spots near the river mouths, which are compensated by losses in the subtropical and tropical oceans. In the Bay of Bengal in particular, NPP is enhanced near the outflow of the Meghna River (up to +36 Tg C yr$^{-1}$), but due to important loss of N further south, total NPP in the basin is lower in all simulations compared to CTR (Table 6).

**Table 4.** Mean annual depth-integrated NPP from model data and observations

| Source | NPP [Pg C yr$^{-1}$] | Description |
|---|---|---|
| UVic CTR | 54.9 | Model |
| UVic NEWS | 55.3 | Model |
| UVic DIN+DON | 55.5 | Model |
| UVic 2xDIN | 55.7 | Model |
| Behrenfeld and Falkowski (1997) | 43.5 | Satellite data |
| Behrenfeld et al. (2005) | 67 | Satellite data |
| Carr et al. (2006) | 51 | Mean of 31 global models |
| Westberry et al. (2008) | 52 | Carbon based, spectral |
| Buitenhuis et al. (2013) | 56 | Model and observational database |

**Table 5.** Minimum and maximum values of NPP difference to CTR simulation as supplement to Figure 12

| Simulation | Difference in NPP [g C m$^{-2}$ yr$^{-1}$] | | | | |
|---|---|---|---|---|---|
| | Bay of Bengal | Yellow Sea | North Sea | Rio de la Plata | Eastern Mediterranean Sea |
| UVic NEWS | -62 | 502 | 246 | 182 | 114 |
| UVic DIN | -69 | 501 | 274 | 258 | 124 |
| UVic 2xDIN | -122 | 544 | 337 | 261 | 130 |

As stated before, decreasing N$_2$ fixation together with higher rates of denitrification partly compensate for the additional N from riverine export and thereby buffer the increase in NPP. Note in this regard that a small fraction of NPP is primary production by diazotrophs, which is lower in the NEWS simulations than in CTR (-0.09 Pg C yr$^{-1}$ for NEWS, not shown here).

In comparison with the NPP pattern in observation (e.g. Behrenfeld and Falkowski, 1997), simulated primary production in
UVic is higher in the tropical upwelling regions along the equator and the northern Indian ocean, while there is too little productivity on the shelves (Keller et al., 2012). In the simulations with riverine N supply, NPP is increased in the coastal regions and, at least for the 2xDIN experiment, lower in the tropical upwelling regions than in CTR.

The increase in NPP is mainly driven by higher rates near the river mouths whereas primary production declines in regions, where rates of N$_2$ fixation are lower as a reaction to the input of riverine N, like in the Gulf of Bengal and near the Amazon
river mouth, but also in parts of the open subtropical and tropical oceans (Fig. 8, Fig. 11 and Fig. 12). This shows also, that the response of ocean biogeochemistry depends on the region where riverine DIN first reaches the ocean.

**Table 6.** Total NPP, $N_2$ fixation and total denitrification from all simulations compared to CTR in some regions with significant decrease or increase in NPP as seen in Fig. 12. NPP is shown in g C $yr^{-1}$, $N_2$ and denitrification in Tg N $yr^{-1}$

| Region | NPP | | | $N_2$ fixation | | | Denitrification | | |
|---|---|---|---|---|---|---|---|---|---|
| | NEWS | DIN+DON | 2xDIN | NEWS | DIN+DON | 2xDIN | NEWS | DIN+DON | 2xDIN |
| Bay of Bengal | -19 | -23 | -31 | -3.5 | -4.2 | -6.5 | +1.1 | +1.4 | +2.4 |
| Yellow Sea | +119 | +133 | +190 | -0.2 | -0.2 | -0.2 | +2.1 | +2.5 | +3.8 |
| North Sea | +35 | +43 | +62 | 0.0 | 0.0 | 0.0 | +0.8 | +1.0 | +1.5 |
| Rio de la Plata river mouth | +17 | +27 | +35 | -6.5 | -1.4 | -2.4 | +0.4 | +0.6 | +0.7 |
| Eastern Mediterranean Sea | +41 | +50 | +53 | -4.0 | -1.4 | -4.4 | +0.2 | +0.2 | +0.2 |

## 3.3 Simulations with regionally activated riverine nitrogen supply

To answer the question which rivers have the highest influence on global marine biogeochemistry, we performed five additional experiments with the same configuration as in the NEWS simulation before, but this time only the rivers of one of five parts of 370 the world transport $NO_3$ to the ocean. The five scenarios simulate the nutrient supply from North American rivers (NAM), South American rivers (SAM), European and Russian rivers (EUR), Asian rivers (ASIA) and African rivers (AFR), respectively.

Total riverine N input varies depending on the rivers. Therefore, the amount of N added to the ocean is different in each of the five scenarios. The highest amount of N is added by Asian rivers (11.7 Tg N $yr^{-1}$). Rivers from South America export 3.5 Tg N $yr^{-1}$, followed by rivers in EUR (3.2 Tg N $yr^{-1}$). The lowest input scenarios are NAM (2.6 Tg N $yr^{-1}$) and AFR (2.3 375 Tg N $yr^{-1}$). The three ocean basins are then affected differently depending on the scenario.

Compared to the control simulation, the differences in $NO_3$ concentrations are small (0-2 mmol / $m^3$), but the patterns differ depending on the origin of $NO_3$ (Fig. 13). It appears that all ocean basins are most affected by rivers in Europe and Russia (EUR) and least affected by rivers from South America (SAM), although rivers from this region provide the second highest N supply to the global oceans. The Atlantic Ocean is most affected by rivers in EUR and to a lesser extend from NAM. Asian 380 rivers lead to a local increase of $NO_3$ concentrations in the Northern Pacific in the upper 2000 m. Here, $NO_3$ is trapped because North Pacific waters upwell in the North Pacific. Extra nitrogen is used by local biota and recycled within the North Pacific. But globally and over all depths, rivers in EUR and to a lesser extent in NAM have the biggest impact on $NO_3$ concentrations in the Pacific. Rivers in SAM slightly decrease $NO_3$ concentrations in large parts of the Pacific Ocean.

In the Indian Ocean basin, $NO_3$ concentrations are higher in the simulations NAM, EUR and AFR. This is because global 385 circulation transports N to remote ocean basins but is not consumed by local primary production nor does it trigger the vicious cycle described before. In contrast, zonally averaged $NO_3$ concentrations are lower in the northern Indian Ocean, if Asian rivers supply N and thereby enhance denitrification (Fig. 13). N from other regions does not trigger the 'vicious cycle' (Landolfi et al., 2013) in the Bay of Bengal, in the model, because it arrives and is used in biological production in other regions, before it can reach the Bay of Bengal.

In all simulations with regionally activated riverine nutrient input, $N_2$ fixation rates are lower than in CTR, with differences

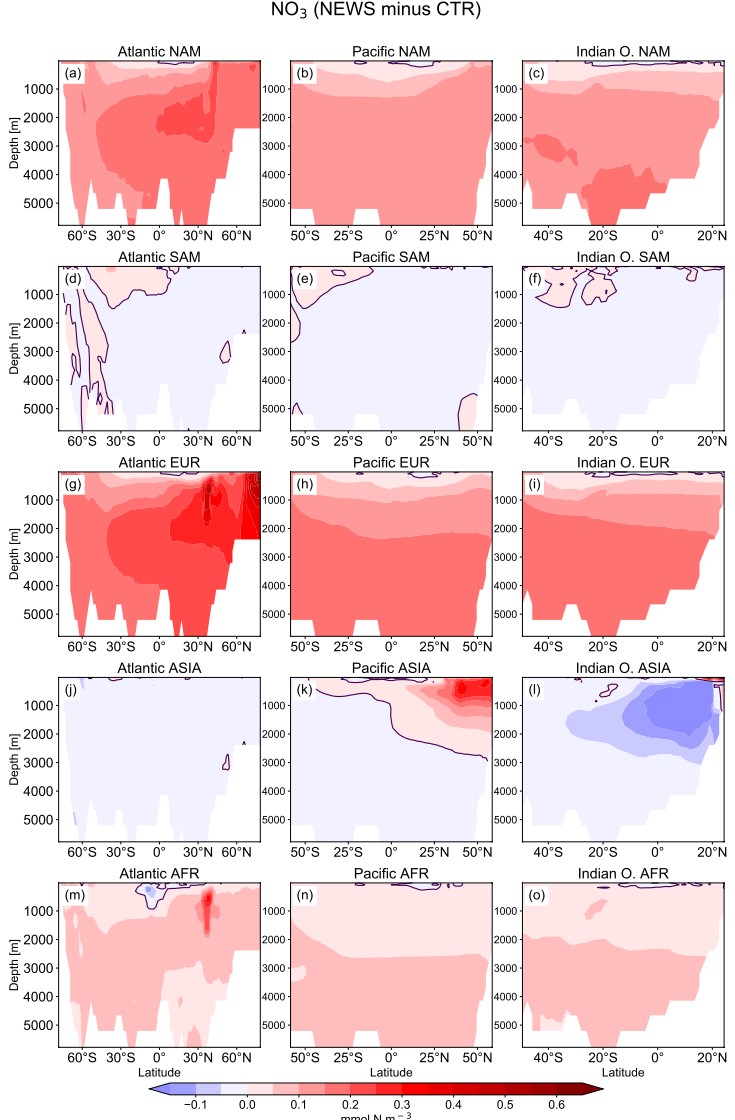

**Figure 13.** Difference of the zonal mean concentrations of NO$_3$ in the main ocean basins (Atlantic, Pacific and Indian Ocean) between regional NEWS and the control simulation (CTR). The regional simulations show export from rivers in North America (a-c), South America (d-f), Europe and Russia (g-i), Asia (j-l) and Africa (m-o), respectively. The violet line indicates the 0.0 mmol N contour line. Maximum differences can be found in the Indian Ocean ASIA, with minimum values at -0.9 mmol N m$^{-3}$) and maximum values at 2.5 mmol N m$^{-3}$.

ranging between -0.8 Tg N yr$^{-1}$ in EUR to -7.7 Tg N yr$^{-1}$ in ASIA (not shown here). This decrease is most prominent in the Bay of Bengal for experiment ASIA, because high DIN export from the Ganges Delta gives the advantage to non-N-fixing phytoplankton, which outcompete diazotrophs.

**Table 7.** Global NPP from different UVic simulations

| Simulation | NPP [Pg C yr$^{-1}$] | Delta NPP per added N [Tg C / Tg N] |
|---|---|---|
| UVic CTR | 54.94 | 0 |
| UVic NEWS | 55.34 | 17.54 |
| AFR | 54.99 | 21.74 |
| ASIA | 55.04 | 8.55 |
| EUR | 55.08 | 43.75 |
| NAM | 54.99 | 19.23 |
| SAM | 54.96 | 5.71 |

Generally, rivers that enter well-oxygenated eutrophic oceans with little N$_2$ fixation have largest impact on the global ocean N inventory. This is especially the case for rivers from EUR and NAM, entering the Atlantic and Arctic oceans at higher northern latitudes. In contrast, the Amazon in SAM is located in an oxygen-deficient region in the tropical Atlantic. The main riverine N supply in ASIA increases N concentrations in the higher northern latitudes of the Pacific, but leads to a net loss of N in the Bay of Bengal.

This has consequences for marine productivity: although NPP is higher in NEWS in most of the coastal oceans, where rivers export DIN, NPP is considerably lower in three regions, where the positive vicious-cycle feedbacks dominate: in AFR in the Gulf of Guinea, in ASIA in the Gulf of Bengal and in SAM, where the river plume of the Amazon river enters the tropical Atlantic Ocean (Fig. 14).

The global NPP rates range from 54.96 Pg C yr$^{-1}$ in SAM to 55.08 Pg C yr$^{-1}$ in the EUR simulation (Table 7). Related to the total amount of N added to the respective simulations via river supply however, EUR contributes considerably more to a widespread increase in NPP, while SAM contributes least (Table 7). ASIA only increases NPP in a confined part of the Pacific Ocean and mainly decreases primary production in other regions like the Bay of Bengal (Fig. 14).

According to our simulations, the ecosystem's response to riverine N supply differs depending on the region of the supply and does not always result in an increase of marine NPP. Note that the sum of the NPP changes in the regional experiments is equal to the change in NEWS, except for parts of the Southern Ocean and the eastern Mediterranean Sea. In the Mediterranean Sea, the sum of the regional NPP changes compared to CTR is higher by 1,8 % than the change in global NEWS. This is due to phosphate limitation in this region in the model, limiting increases in NPP in the NEWS simulation.

## 4 Limitations and further discussion

The results of the simulations with UVic and riverine N have to be evaluated in the context of previous studies. Lacroix et al. (2020) for example found that adding riverine nutrient supply increased primary production essentially in some "hot-spots"

NPP (NEWS minus CTR)

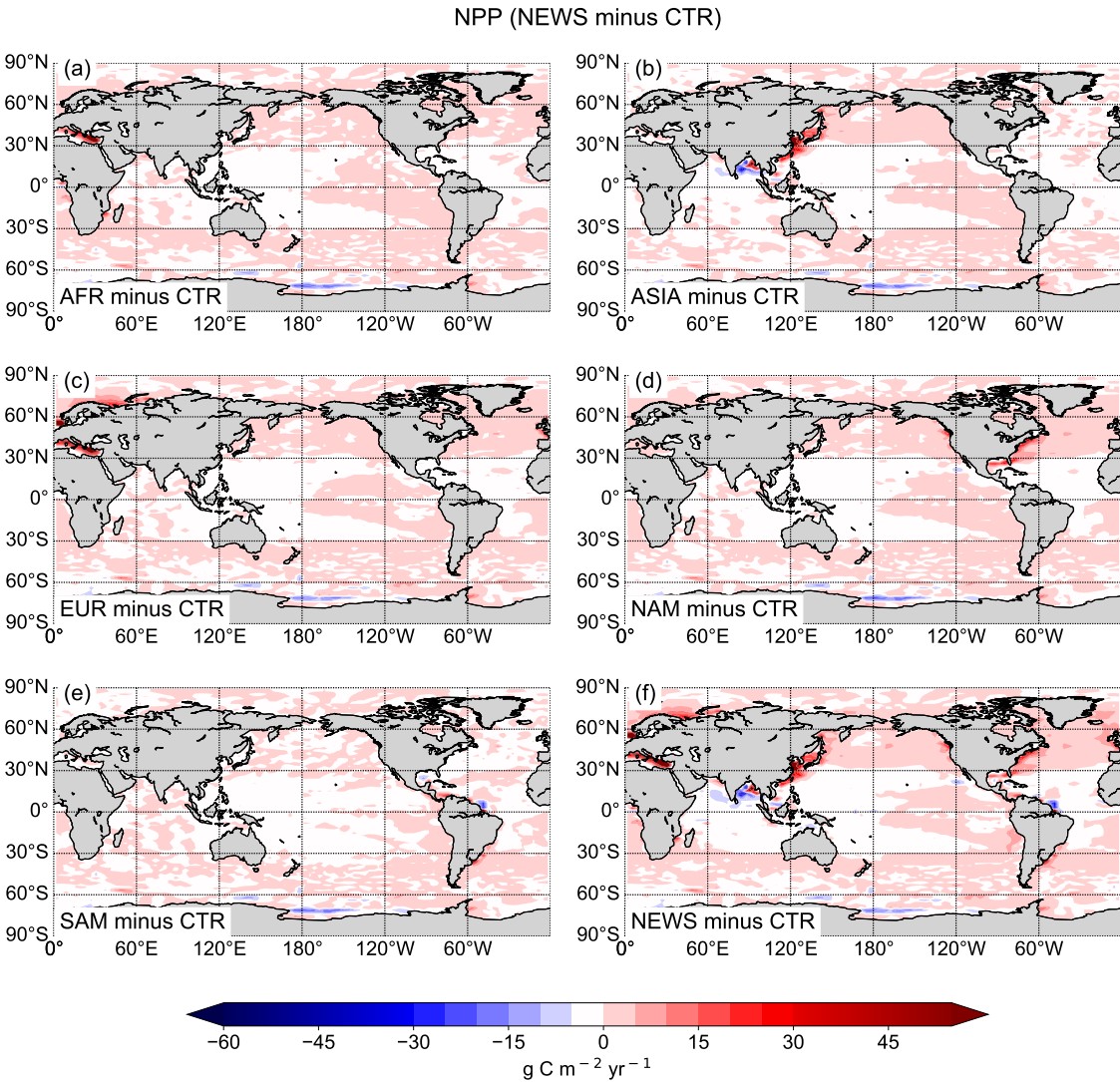

**Figure 14.** Difference of the global vertically integrated rates of primary production (NPP) between regional NEWS and the control simulation (CTR). The regional simulations show export from rivers in Africa (a), Asia (b), Europe and Russia (c), North America (d) and South America (e).

near the river mouths. While we have observed a qualitatively similar phenomenon in our simulations, these hot-spots differ between both studies. This is due partly to the coarser resolution of UVic. The grid configuration used by Lacroix et al. (2020) (GR15) consists of a bipolar grid which resolves the ocean horizontally at around 1.5° and through 40 unevenly spaced vertical layers. Although riverine N is simulated as fixed percentage of P, dynamic nitrogen fixation by cyanobacteria is included as well

as nitrogen deposition and denitrification. From the semi-enclosed seas, which present higher NPP in the study by Lacroix et al.
(2020), only the Yellow Sea is adequately resolved in UVic and shows also a significant increase in NPP. But the patterns of primary production differ in several other aspects. Where NPP is substantially increased in the subtropical and tropical eastern Pacific in Lacroix et al. (2020), there is hardly any change in UVic-NEWS. In the Bay of Bengal, where we found decreased NPP upon addition of riverine N supply in UVic, their model simulates an increase. The main reason for these differences is the fact, that Lacroix et al. (2020) included more than just N from river discharge. Especially the supply in additional phosphate plays an important role for NPP, and is for example particularly high in the Bay of Bengal in their simulation. Furthermore, the magnitudes of oceanic nutrient inputs do not differ substantially between the two simulations analyzed in the study of Lacroix et al. (2020): the total N input is 25.2 Tg N yr$^{-1}$ for the reference simulation (REF) and 27.0 Tg N yr$^{-1}$ for the simulation with riverine nutrient supply (RIV). The reference distributions of NPP (REF) in Lacroix et al. (2020) also differs with regard to UVic. NPP from UVic is notably higher in the Indian Ocean and the western tropical Atlantic, but lower in the Southern Ocean. These absolute higher values of NPP in the open oceans can be accounted for by the parameterisation of NPP in our model, where open oceans have to compensate for the lack of higher coastal production in order to achieve estimated annual NPP within the range of 51 to 67 Pg C yr$^{-1}$, that include coastal NPP (Keller et al., 2012). But as both reference distributions of NPP differ (from Lacroix et al. (2020) and CTR), it is no surprise, that NPP also presents different patterns in the simulations with riverine nutrient supply (RIV and NEWS, respectively).

Riverine nutrients only reach the ocean in very localized areas. In our simulation with the NEWS data set from Mayorga et al. (2010), we overestimate the effects of adding N from river discharge, because DIN is exported directly from the river mouth to the ocean, as our global model does not fully resolve shelf seas and coastal oceans. In reality, part of these nutrients stays on the shelf or is buried or denitrified in coastal sediments. We also do not account for the buffer effect of the coasts, that could be parameterised, as shown by Sharples et al. (2017) and Izett and Fennel (2018). Nevertheless, even without taking these trapping processes into account, the biogeochemical feedbacks of the ocean buffer higher increases in N concentrations. The absolute increase in marine primary production is small (between +0.7 % in NEWS and +1.3 % in 2xDIN). However, relative to the amount of N added to the global ocean, primary production increases yearly by 17.5 Tg C per additional Tg N in NEWS (16.0 Tg C per Tg N in 2xDIN). As we have shown, primary production increases mainly near the river mouths, where high nutrient loads are injected in shallow ocean areas, creating production "hot spots", while only small changes in production have been found in the open ocean.

Other studies with additional N supply also found only moderate increase in global primary production rates. Da Cunha et al. (2007) for example predicts increases in NPP up to +5 % for the global ocean, but using a high DIN scenario which includes 7.1 Tmol N yr$^{-1}$ (corresponding to $\sim$ 100 Tg N yr$^{-1}$). Da Cunha et al. (2007) also include silicate, iron and dissolved inorganic carbon, but concluded that riverine N may have the higher impact on primary production.

Like Somes et al. (2016), who also simulated a very small increase in NPP upon the addition of N deposition in their model, we found that decreasing N$_2$ fixation and increasing denitrification act globally as negative feedbacks. Compared to the total amount of N added by the rivers at the end of the simulation, only 2.3 % (NEWS) to 2.6 % (2xDIN) is retained in the global inventory. The feedbacks compensate for much of the nitrogen addition. In regions of low oxygen concentrations, these

feedbacks even overcompensate the external perturbation in terms of riverine N supply, by forming a "vicious cycle" (Landolfi
et al., 2013), consuming more N than provided by the rivers. This is especially the case in the Bay of Bengal. However, we
are aware of the fact that UVic, like several other models, currently misplaces the oxygen minimum zone from the Arabian
Sea to the Bay of Bengal. It is likely, that N supply by Asian rivers would lead to a somewhat larger increase in the oceanic N
inventory, if the high nutrient input from the Ganges Delta would not meet the high denitrification zone in the Indian Ocean
(Johnson et al., 2019).

Including other nutrients in addition to N, especially P, could change the setting, especially in regions that are phosphate
limited. While this is the logical continuation of the current study, the scope of this project was to explore the consequences of
locally high N injections on the N cycle and its feedbacks. Furthermore, as rivers supply relatively more N than P to the global
ocean, excess N would still be supplied to the coastal oceans (Turner et al., 2003).

## 5   Conclusion

In this study a new model component was added to the global Earth System Model UVic, simulating DIN supply from river
discharge. At the end of the 10000 years of simulations, the N budget has reached a new steady state. The main conclusions
regarding N cycle and marine productivity include:

– that riverine N added to the coastal ocean is taken up by near-coastal biology but also exported in the deeper ocean and
  circulated worldwide.

– Despite the continuous addition of N to the system, global marine N concentrations and marine productivity do not
  increase substantially (+1.12 % N and +0.72 % for NPP) in our simulations.

– In the coastal regions and especially in some hot-spot regions near the river mouths, riverine nitrogen input leads to
  higher primary production. Globally, NPP rates increase up to 17.5 Tg C/yr per Tg N/yr added to the ocean.

– The globally negative feedbacks of the N cycle buffer most of the increase in $NO_3$ concentrations and in NPP. $N_2$ fixation
  decreases promptly after the beginning of the simulations, partly compensating for the additional N at the surface of the
  ocean, likewise to the N deposition experiments by Somes et al. (2016) and Landolfi et al. (2017). Water column and
  benthic denitrification are higher compared to a control simulation without riverine N input and play an important role
  in low-oxygen regions that, moreover tend to expand upon the addition of riverine N supply and generate a net N loss.

– In our regional simulations we have shown, that NPP can even decrease locally depending on the region where N reaches
  the ocean. While N from river discharges from North America and Europe (and Russia) is also circulated and exported to
  the deeper ocean, N from Asian rivers is trapped in the western Pacific or even partly lost via denitrification in oxygen-
  deficient regions, like it is the case in the modeled Bay of Bengal.

– The biogeochemical feedbacks of the ocean buffer further increases in global N concentrations and global NPP. Hence,
  the result suggests also, that ocean fertilization with nitrogen alone (as proposed for example by Harrison (2017)) may

not have the desired effect. Indeed, simulated carbon export, evaluated at the 122 m level and including all the shelf regions, increases globally by only 0.06 Pg C yr$^{-1}$ in our NEWS simulation, representing less than 10 % of the amount estimated by Harrison (2017) to be the upper limit of sequestered carbon in the ocean from on-going fertilization with nitrogen. On short time scales target N fertilization might have some impact, but once the vicious cycle has a chance to start, then the findings would probably be the same, as shown e.g. by Somes et al. (2016). But further research would need to be done for targeted spatial and temporal N additions at different levels in N limited regions.

We have found, that likewise to atmospheric deposition, river supply of nitrogen is not only relevant for the coastal system but also for marine biology in the global ocean. But while atmospheric deposition provides only N, rivers supply also P to the ocean. Adding P in addition to N in the coastal oceans may change the response of the ecosystem, especially if N limitation is overcome. Tyrrell (1999) stated, that nitrate is the "proximate limiting" nutrient in surface waters, the most limiting nutrient to instantaneous growth. Including phosphate, as the "ultimately limiting nutrient", could change our story on longer time scales. Future research will therefore include model experiments with the combination of riverine nitrogen and phosphorus.

*Code and data availability.* The model code we used as well as input and output data are available online (Tivig, Miriam, Keller, David P., and Oschlies, Andreas (2020). Supplementary Data to "Feedbacks in the marine nitrogen cycle limit the impact of riverine nitrogen supply on global marine biology and biogeochemistry in an Earth System Model ", https://data.geomar.de/thredds/20.500.12085/59977a36-e8e7-4348-a4e8-2b13f3913590/catalog.html). More information on the original NEWS2 data set is available from the Global NEWS group at the web site http://icr.ioc-unesco.org/index.php?option=com_content&view=article&id=45&Itemid=100002. Please email Emilio Mayorga at mayorga@marine.rutgers.edu to obtain this data set.

*Author contributions.* MT developed the research concept in discussion with AO and DPK. DPK provided the initial model code, which was further developed, run, and analysed by MT. MT analysed the model output and visualised the results. MT wrote the manuscript with contributions from all co-authors.

*Competing interests.* The authors declare that they have no conflict of interest.

*Acknowledgements.* We gratefully acknowledge E. Mayorga, S. Seitzinger and their co-authors for making their database of Global Nutrient Export from WaterSheds 2 (NEWS2) available for our study. AO and DPK acknowledge funding from the European Union's Horizon 2020 Research and Innovation Program under grant 820989 (project COMFORT, "Our common future ocean in the Earth system — quantifying coupled cycles of carbon, oxygen, and nutrients for determining and achieving safe operating spaces with respect to tipping points") and OceanNETs (grant #869357). The work reflects only the author's view; the European Commission and their executive agency are not responsible for any use that may be made of the information the work contains. This work was also supported by the German Research Foundation

(DFG) as part of the research project SFB 754 "Climate-Biogeochemistry Interactions in the Tropical Ocean". We would also thank the GEOMAR's Biogeochemical Modelling group for many fruitful discussions.

We thank the two anonymous referees for their very helpful and constructive comments and suggestions.

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
