# Peer review of "Riverine nitrogen supply to the global ocean and its limited impact on global marine primary production: a feedback study using an Earth System Model"

_Biogeosciences, 2021_

## Author Comment (AC1)

**Comment on bg-2021-101**
**Anonymous Referee #1**

Referee comment on "Riverine nitrogen supply to the global ocean and its limited impact on global marine primary production: a feedback study using an Earth System Model" by Miriam Tivig et al., Biogeosciences Discuss., https://doi.org/10.5194/bg-2021-101-RC1, 2021

This is an interesting article exploring the long-term impact of riverine nitrogen inputs on the global ocean nitrogen inventory and associated primary production. The authors show that in simulations that have reached equilibrium, impacts on the global N inventory and primary production are highly limited due to feedbacks on N fixation and denitrification. I appreciated the candidness of the discussion on the modelling approach's strengths/weaknesses and how the results differ from those previously published in this area. The main discussion that I think might be expanded upon is how indicative simulations that have reached equilibrium are for policy relevant timescales (interannual to decadal). **Put another way, how do the authors' main conclusions change over the course of these 10000-year simulations? Are the implications different for watersheds that are experiencing rapid increases or decreases in nitrogen export at present?** Most of my comments and suggestions relate to how the manuscript text could be better structured and the figures could be made much easier to interpret. Subject to these changes, I would be happy to recommend for publication.

→ Thank you for your recommendation and positive feedback.
Regarding your question, how the main conclusions change over the whole simulation, we will include additional text in the revised manuscript, indicating that the main changes in the global marine nitrogen inventory appear in the first 4000 years of the simulation. After this time, the global nitrogen budget is almost in equilibrium. After the perturbation by additional nitrogen input via the rivers, marine primary production increases globally in all our simulations, but only for less than 2000 years. The most prominent changes have been found in the regions, where rivers currently export higher amounts of nitrogen, like for example in the East China Sea.

**Minor comments**

L23. "atmospheric" –should probably be dissolved/aqueous.
→ Biological dinitrogen ($N_2$) fixation refers here to atmospheric $N_2$, dissolved in the ocean, which is reduced to ammonia

L26. Aren't these "model concepts" observationally /experimentally derived? Some reference to the empirical evidence would be useful here. Maybe also its limitations if it's heavily based on given species (eg Trichodesmium).
→ We will include more references, like:
- Karl et al. (2002): Dinitrogen fixation in the world's oceans, Biogeochemistry 57/58: 47–98.

- Landolfi et al. (2018) Global Marine N2 Fixation Estimates: From Observations to Models.Front. Microbiol. 9:2112. doi: 10.3389/fmicb.2018.02112
- Zehr and Capone (2020): Changing perspectives in marine nitrogen fixation, Science 15 May 2020, Vol. 368, Issue 6492, eaay9514, DOI: 10.1126/science.aay9514

and indicate that a part of our knowledge of N2-fixation is still based on the original limited assumptions based on given species especially Trichodesmium.

L31. concentrations "of" fixed N → will be corrected

L32. "the consumption of O2"- Do you mean the consumption of O2 during remineralisation? If so, be explicit. → "heterotrophic O2 consumption during organic matter remineralization"

L47. The Séférian et al. 2020 reference is perhaps worth citing here as it summarises the inclusion of riverine inputs in recent models. → This reference will be included

L51. "real" should probably be realistic or observed. → "without observed nutrient fluxes"

L59. Maybe clarify what is meant by N export here. Riverine delivery? For many in the ocean biogeochemistry community export is instinctively a vertical flux. → "Their focus was on pre-industrial nutrient input from the rivers"

Figure 1. More detail is needed on the tracers in the figure legend. → Yes, it will be included in the caption in the revised manuscript. This text will read, "Ecosystem model schematics for the NPZD model with the prognostic variables (in square boxes) and the fluxes of material between them, indicated by arrows. The prognostic variables include two nutrients, nitrate ($NO_3$) and phosphate ($PO_4$), two phytoplankton (nitrogen fixers $P_D$ and other phytoplankton $P_O$) as well as zooplankton (Z), sinking detritus (D), and dissolved oxygen ($O_2$). nitrate ($NO_3$) and phosphate ($PO_4$) are linked through exchanges with the biological variables by constant (Redfield) stoichiometry."

L100. "atmospheric" –should probably be dissolved/aqueous. → "Nitrogen gas dissolved in seawater "

L101. Are they limited by a max NO3 concentration? I know much of this will be in the cited references but more detail is required on N-fixation in the model. Highlight perhaps that most models don't have explicit diazatrophs and this is an advantage of using uvic. What is the diazatroph PFT based on? How does N-fixation compare to observations where they exist?

→ We will include a statement like this in the revised manuscript: "Diazotrophs are not limited by NO3 concentrations, nor by a maximum NO3 concentration. The explicit integration of diazotrophs permitting the estimation of nitrogen fixation, is not given in all ocean models but makes UVic a good choice to study nitrogen cycle feedbacks. The maximum potential growth rate of diazotrophs is not only based on temperature as in most models, but also on dissolved iron, which is necessary e.g. for photosynthesis or the reduction of nitrate to ammonium (Keller et al., 2012; Galbraith et al, 2010). Keller et al. (2012) found that the global nitrate fixation rates were within the range of global nitrate fixation rates from estimations and the patterns of $N_2$ fixation from the new model were mostly consistent with observations, as far as they are known (Sohm et al. 2011).

L105-110. As with the above comment, some comparison on how denitrification in the model compares to observations would be very useful. I think a global map of N-fixation and denitrification in the model CTR is required.

→ In the revised manuscript we will include some more maps with global distribution of denitrification and N$_2$-fixation.

L140. It should be made clearer on first use that NEWS etc are simulation names. → We will include a sentence about the simulation names at this place.

Table 1. For clarity I would remove UVic from the simulation names as this is not repeated in the main text. I would also add a CTR row. → Yes, absolutely right. This has been done for the revised manuscript.

L156. "vary a little" – please quantify this → "Global average NO3 concentrations only vary by 1-5 mmol m$^{-3}$ between the simulations

L151. This should really be called a "Results and Discussion" section. → Yes, some of the final discussions have been directly included in the results chapter, so we will rename this section as you suggest.

L160. The wording here needs to be clearer. "At smaller scales…globally higher." This reads like it contradicts L167-168. → In the revised manuscript we change this sentence to "Nevertheless, in all three simulations (NEWS, DIN+DON, and 2xDIN), NO3 concentrations are globally higher compared to the control simulation (CTR)…"

Figure 3. Axes are missing labels and units here. → This has been corrected in the revised Figure

Figure 5. I find it very difficult to see differences between positive and negative anomalies using this colorbar. I suggest changing to something far more distinctive (e.g. red for negative anomalies). The same applies to other figures using this scale. → All Figures concerned have been updated with a new colorbar (in blue and red). We have chosen a delta colorbar with blue for negative and red for positive values.

L175-177. This is difficult to see in Figure 5 maybe cite figure 6 here. → Citation of Figure 6 included here.

Figure 6. The depths given in the figure don't match the legend. → Figure 6 has been updated for the revised manuscript.

L181-183. This sentence is confusing and needs rephrasing.
→ We have improved the text to read: "But in the deeper northern Indian Ocean basin down to approximately
2000 m, NO3 concentrations are significantly lower in NEWS than in CTR. Considering the zonal average of the Indian Ocean, NO3 concentrations are lower by -0.7 to -0.9 mmol N m$^{-3}$, and even more if only the zonal average of the Bay of Bengal is considered."

Figure 7. Label missing from panel c. → Figure 7 has been updated for the revised manuscript.

L187. typo. "amounts to an increase of only 1.1…" → corrected

L209. I'm not sure the language here is accurate. Presumably the model is not explicitly trying to compensate anything. Wouldn't this be better described as enhanced denitrification sinks promoting conditions that favour N-fixers over the other PFT type and consequently global N-fixation rates are higher?

*"To compensate for this additional N sink, the model estimates higher fixation rates."* changed to:

→ "The additional N sink in form of benthic denitrification promotes conditions that favor N-fixers, i.e., diazotrophs, leading to higher nitrogen fixation rates."

L210-213. The discussion of other literature here before properly explaining your model results is confusing. Where these papers have used the same model this should be clear.
→ We will try to avoid confusion here and change the paragraph to: "Previous studies with UVic have shown, … (Somes et al., 2016)" The other studies cited here have shown similar results but with other models, so we will come to them later in the discussion.

L216. "where NO3 concentrations are substantially reduced relative to the CTR" → The text has been updated, thanks for the rephrasing.

Figure 8- See earlier comments on how it would be nice to see global Nfix in the control.
→ Thanks for the suggestion, this has been done in the revised Figure.

Figure 9. Suggest using different color palettes for mean states and anomalies. → Thanks for the suggestion, this has been done in the revised Figure (see also response to Figure 5)

L219-227. The balance between results of the model simulations and the discussion of other literature needs to be more organised. The presentation of discussion before results is quite confusing.
→ To address this critique, the whole paragraph (previously l. 190-241) has been restructured and divided in two subsections: "3.1.2. Denitrification and nitrogen fixation" and "3.1.3 The N-cycle feedback mechanisms"). (See whole paragraph at the end of this document)

L221-222. This is a bit rushed and therefore confusing. I think more detail is needed here on this mechanism, the difference between the stoichiometry of N-fixation and denitrification and how spatial and temporal coupling is important for the positive feedback to occur.
→ Sorry that this was confusing we have updated the text to read, "This is due to the stoichiometry imbalance created by the combination of these processes. Denitrification occurs in anoxic or suboxic environments, where nitrate or nitrite can be used as a substitute terminal electron acceptor instead of oxygen. Denitrification consumes 7 mol of NO3 for every mole of organic N provided by N2-fixation, when these processes equilibrate with each other over long timescales. This imbalance generates a net loss of fixed N, even if N is continuously added via N2 fixation."

Table 3. Benthic denitrification appears to be twice the magnitude of that in the water column. This doesn't seem to be reported and discussed in the manuscript. Does this have

implications for models lacking benthic denitrification?

→ I don't see, where benthic denitrification appears to be twice the amount of water column denitrification (WCD), but indeed benthic denitrifications (BD) is an important process (e.g. Somes et al., 2016; Somes et al., 2013) and for both global estimates vary considerable: for WCD estimates are between 50 and 150 Tg N yr-1, for BD between 100 and 300 Tg N yr-1 (Galloway et al., 2004; Gruber, 2004; Bohlen et al., 2012; Somes et al., 2013). In the revised manuscript we will address this point. Our global results for N2 fixation, WCD and BD stay in the range assumed for a balanced fixed-N budget in the preindustrial ocean e.g. by Somes et al. (2013). BD is more evenly distributed than WCD. Therefore, to study regional effects, it is helpful to include both processes. Nevertheless, models can have a balanced nitrogen budget without benthic denitrification.

L246. "…and vary little between…" → corrected

L251. "smaller scales" is ambiguous. Here and elsewhere I recommend being more specific e.g basin/watershed/coastal scales etc.
→ Yes, at this place we changed the vague formulation to "Nevertheless, with rivers supplying N to the ocean, differences are visible at coastal scales: NPP increases locally, close to the river mouths

L257-259. Differences in spatial patterns between these simulations are difficult to discern it looks more like the magnitude of change is the only difference. → We will include this remark to the text: "The main differences are in the magnitude of NPP, however, some regions with higher NPP can only be found in the simulation 2xDIN in the open ocean basins." Furthermore, we will change the colorbar of the figure in blue and red like stated for Figure 5, in order to make the differences clearer.

L259. Are these subtropical and tropical regions where N-fixers are predominately confined to?
→ Yes, these are quite exactly the regions, where in UVic diazotrophs can be found. We will include a comment on this in the manuscript and refer to the new plot on figure 8.

L297. Maybe "exported again" should be "recycled" here. → yes. "recycled" is what was meant.

L301. And presumably not all the N can be consumed via local primary production due to other constraints.
→ Yes, we will include: "N is transported to the Indian basin but is not consumed by local primary production nor does it trigger the vicious cycle described before".

Table 6. Maybe the increase in NPP per quantity of additional N would be a useful metric to add to this table given each watershed provides different total N delivery.
→ Yes, this is a good idea. This has been calculated already and can be easily added to the existing table.

L325. For clarity I think "inhibiting additional NPP" should be "limiting increases in NPP".
→ Thanks for your suggestion, it will be changed.

Section 4. Given the extensive discussion of the Lacroix et al. 2020 paper, I think a few more details are required to compare the studies properly. What was their model resolution, did they have explicit N-fixers and benthic denitrification?

→ We will include this information: "This is due partly to the coarser resolution of UVic. The grid configuration used by Lacroix et al. (2020) (GR15) consists of a bipolar grid which resolves the ocean horizontally at around 1.5° and through 40 unevenly spaced vertical layers. Although N is simulated as fixed percentage of P, dynamic nitrogen fixation by cyanobacteria is included as well as nitrogen deposition and denitrification."

L358. The emphasis here and in the conclusions (L376) doesn't really match the findings. I would say the feedbacks do much more than "partly compensate" the riverine fluxes. Maybe this would be clearer if you gave the % of added N that is retained in the inventory at equilibrium or some other metric of feedback strength.

→ Yes, you are right. We will add: "Compared to the total amount of N added by the rivers at the end of the simulation, only 2,3 % (NEWS) to 2,6 % (2xDIN) is retained in the global inventory. The feedbacks compensate for much of the nitrogen addition and in some regions even overcompensate them."
The sentences in the conclusion were formulated this way, because some of the additional N is still accumulated in the deeper ocean.

L388-389. I'm not sure this "upper limit" conclusion would hold if N fertilisation were targeted spatially and temporally in regions of N limitation. Perhaps this should be toned down a little.

[revised manuscript text omitted]

Reductions in N2 fixation can then partly explain the lower NO3 concentrations at the surface of the tropical and subtropical oceans in NEWS, even though these areas are far from riverine N input (see 3.1.1. and Fig. 6 first row). We remind here that these results show the distribution at steady state after 10 000 years of riverine nitrogen supply. Not all fixed nitrogen is consumed by biological activity, but part of the additional N is also transported with ocean circulation and can "replace" N from nitrogen fixation in regions far off the coast, leading to decreasing N2 fixation at the surface of the tropical and subtropical oceans.

**3.1.3. The N-cycle feedback mechanisms**

The interaction between the mechanisms described just before, N2 fixation, denitrification, and riverine nitrogen supply, can also explain the significant loss in NO3 in some regions localized before: the Gulf of Guinea, the Gulf of Bengal and the western coast of Central America (Fig. 5 and Fig. 7). In addition, these three regions have also in common that they are known to have very low oxygen concentrations. In the Bay of Bengal, oxygen concentrations even though higher at the surface in NEWS than in CTR, are very low in the NEWS simulations in the subsurface waters and the whole deeper basin (Fig. 7). These suboxic waters are furthermore located in proximity to riverine N input and high denitrification rates (Fig. 9). While total denitrification rates (benthic and water column denitrification) are already quite high in CTR, they are further increased in NEWS, DIN+DON and 2xDIN in the northern Bay of Bengal, adjacent to the river delta.

Landolfi et al. (2013) found that the negative feedback mechanism between N2 fixation and denitrification, generally stabilizing the marine N inventory, can turn into a destabilizing positive feedback, generating runaway N loss, if a close spatial association of N2 fixation and denitrification occurs. This is due to the stoichiometry imbalance created by the combination of these processes. Denitrification occurs in anoxic or suboxic environments, where nitrate or nitrite can be used as a substitute terminal electron acceptor instead of oxygen. Denitrification consumes 7 mol of NO3 for every mole of organic N provided by N2-fixation, when these processes equilibrate with each other over long timescales. This imbalance generates a net loss of fixed N, even if N is continuingly added via N2 fixation.

The 'vicious cycle' described by Landolfi et al. (2013) is triggered in the Bay of Bengal by the input of new N from riverine export near oxygen minimum zones, explaining the NO3 deficit

found in the simulated Bay of Bengal (Fig. 5). Note that UVic, similar to most other biogoechemical ocean models, misplaces the main oxygen minimum zone from the Arabian Sea to the Gulf of Bengal (Séférian et al., 2020). In reality, high water column denitrification has been observed in the Arabian Sea, while in the Gulf of Bengal highly variable oxygen concentrations seem to inhibit denitrification (Johnson et al., 2019; Bange et al., 2005).

At the end of the simulation, the global marine N inventory is higher by 5278 Tg N in NEWS compared to CTR, which corresponds to 1.12 % of the global N inventory in CTR and 2.3 % of the total riverine N input over the 10000 years of the simulation. Even for the highest scenario (2xDIN), the total increase in global N represents only +2.53 % of the reference N inventory. Most of the additional N input through river discharge is thus compensated for by the feedbacks of the N cycle.

However, relative to the total additional input, the N increase in 2xDIN is higher than in NEWS (+2.6 % compared to +2.3 %), which means that the negative feedbacks do not compensate in 2xDIN as much as in NEWS. A possible reason for this result could be, that the main negative feedbacks, resulting in loss of N, take place in very localized low-oxygen areas, that cannot expand further (e.g. Bay of Bengal), while riverine N is supplied through river mouths scattered over the world.
* * *
**Additional literature**

Capone, D. G., Burns, J. A., Montoya, J. P., Subramaniam, A., Mahaffey, C., Gunderson, T., Michaels, A. F., and Carpenter, E. J.: Nitrogen fixation by Trichodesmium spp.: An important source of new nitrogen to the tropical and subtropical North Atlantic Ocean, Global Biogeochem. Cy., 19, GB2024, doi:10.1029/2004GB002331, 2005.

Galbraith, E. D., Gnanadesikan, A., Dunne, J. P., and Hiscock, M. R.: Regional impacts of iron-light colimitation in a global biogeochemical model, Biogeosciences, 7, 1043–1064, doi:10.5194/bg-7-1043-2010, 2010

Sohm J. A., Webb E. A., and Capone D. G.: Emerging patterns of marine nitrogen fixation, Nature Reviews Microbiology, 9, 499-508, 2011.

Somes, C. J., Schmittner, A., and Altabet, M. A.: Nitrogen isotope simulations show the importance of atmospheric iron deposition for nitrogen fixation across the Pacific Ocean, Geophys. Res. Lett., 37, L23605, doi:10.1029/2010GL044537, 2010a.

---

## Author Comment (AC2)

**Comment on bg-2021-101**
**Anonymous Referee #2**

**Major criticisms:**
- the model description has a few gaps that might benefit from filling and improved clarity
- the experimental design involves several highly idealised simulations and it could be clearer which hypotheses are being tested when these are being framed; for instance, rather than use a single experiment that scales river inputs, a suite of scaling runs could instead assess the strength and saturation of feedbacks
- the results are generally clear enough, but there are omissions or odd choices in the results presented; it would be helpful, for instance, to have tables which bring together the major N-cycle processes across the different runs

RESPONSE

Thank you for the comments and suggestions.
In the revised manuscript we address the major criticisms as followed:
- we have included more details in the model description and included the full description of the NPZD model as shown in Figure 1.
- In the revised manuscript we will add the hypotheses to the description of the experimental design. While the first run, where we add the riverine DIN from the NEWS model is a test run, to evaluate first, how the model reacts to the riverine nutrient input and second how the nitrogen cycle responds to it, the following simulations, where we increased the amount of nitrogen added to the ocean, are already scaling experiments to test the mechanisms and feedbacks described for example by Landolfi et al. (2017) and Somes et al. (2016).
  We do not have the capacity to include new model runs in this study, but for a following study we are considering doing some more explicit scaling runs.
- Table 3 in the current manuscript brings together river supply in N, global N2 fixation, global benthic and global water column denitrification for the simulations CTR, NEWS, DIN+DON and 2xDIN. We can provide this table also for the simulations with riverine nitrogen supply from different regions.

**Minor comments:**

Pg. 1, ln. 1: I might be inclined to make a distinction between the natural and the anthropogenic nitrogen cycles; we have so radically modified the N-cycle that negative feedbacks may have been swamped; in any case, definitely: make it clear in the abstract whether you consider that you are dealing with the natural or modern N-cycle
→ Yes, that is correct. We mention it in the revised abstract and in the main text, that we consider the natural N-cycle. Nevertheless, as stated in line 118, the NEWS dataset is based on river supply in the year 2000 and also includes natural and anthropogenic biogeophysical properties. See also response to next comment.

Pg. 1, ln. 18: "steady state" - I'm not enough of an expert in riverine supply to be sure, but my first reaction is that the scale of anthropogenic inputs of nitrogen to the ocean must make this assumption questionable. It's an assumption I'm happy to make in my own tangential work for simplicity, but where a study is addressing it head-on, I'd expect something on the anthro perturbation to the N-cycle

→ We will address this very relevant remark in the introduction starting l. 44:
"At this place it is appropriate to include some remarks about the anthropogenic perturbation of the N-cycle. In the Anthropocene, human activities have led to increased inputs of fixed nitrogen from the atmosphere and through different sources of runoff from land (e.g. Somes et al., 2016; Kim et al., 2014, Lamarque et al.,2013) which also indirectly impacts $N_2$ fixation (e.g. Krishnamurthy et al., 2006). At the same time, warming and deoxygenation can lead to increased N loss (Oschlies et al., 2019). The question has been therefore raised, if the global nitrogen budget could still be considered to be in steady state. While these combined effects on the nitrogen budget are still very uncertain, some studies suggest, that imbalances could be limited due to internal feedbacks of the N cycle (Landolfi et al., 2017; Somes et al., 2016; Krishnamurthy et al., 2006)."

Pg. 2, ln. 25: "slowly" - to assist less familiar readers, please expand on why we might expect (or why we know) diazotrophs to be slow-growing

→ New text has been added stating that: "Previous studies have shown, that at least the most common diazotrophs, especially Trichodesmium, have low growth rates relative to many non fixing phytoplankton (e.g. Capone et al., 1997). Indeed, while these organisms are able to fix N2 when reactive nitrogen is scarce, this turns into an disadvantage in regions where N is more abundant, because N2-fixation requires more energy (Tyrell, 1999). The slowly growing diazotrophs are then, rapidly out-competed by non-fixing organisms, if enough P and other nutrients are present (Tyrrell, 1999). Note that part of our knowledge of N2-fixation and most modern models concepts are still based on the original limited assumptions based on given species especially Trichodesmium."

Pg. 2, ln. 29: "most marine organisms" - except diazotrophs, of course
→ "for most other marine organisms"
Pg. 2, ln. 32: "consumption of O2" - this statement is perhaps confusing as people will be aware that growth of phytoplankton *produces* oxygen; I know what you mean here, but others might not
→ Thanks for your remark, we will change the text to: "heterotrophic O2 consumption during organic matter remineralization"
Pg. 2, ln. 34: "work together" - do they really "work together"?; might it not be fairer to say that they work independently, but between them the nitrogen cycle is balanced (which it inevitably must be)
→ Text updated to read: "These two processes, N2 fixation and denitrification, both contribute to regulate the global marine N budget."
Pg. 2, ln. 38: "considered in this study" - it might be useful to indicate the size of the deposition sink so that it's clear why it's ignored here; also, my first thought here was that you didn't want to have to add an additional low importance process, but it looks like someone has already done this for your model; so perhaps be very clear here why you're ignoring it

→ Thanks for your comment. We will address this and delete the sentence in l.38, but add some more explanations in l.68-75 (see comment to that line) ".

Pg. 2, ln. 39: "highly" - how high is "highly"? do you have an estimate of how much the riverine flux has been affected by human activities, or is this still highly uncertain?

→ We will tone down this somewhat and include: "Although riverine N is not the main source of N for most marine environments, it can become important, as it is directly influenced by human activities. Seitzinger et al. (2010) e.g. estimated that global nitrogen export by rivers to the coastal waters increased by 17,7 % from 1970 to 2000. Nitrogen is known to impact…"

Pg. 2, ln. 47: "2008" - 2008 doesn't seem very "current"; perhaps reword or find a better Example → "However, often global biogeochemical ocean models still omit…"

Pg. 2, ln. 50: "as on" - "as on" -> "*than on* the actual quantity of nitrogen nutrient" → thank you, we have corrected this.

Pg. 2, ln. 55: "long enough" - you should say what you think the relevant time period is; I believe Tyrrell (1999) puts an estimate on this based on input flux and ocean inventory

→ "likely not long enough to study the feedbacks of the nitrogen cycle in the open ocean, considering that the mean residence time of fixed nitrogen in the ocean has been estimated to be a few thousand years (Gruber, 2004)."

Pg. 2, ln. 57: "estimations of" -> "estimated" → will be corrected

Pg. 3, ln. 62: "EMICs" - a passing mention of computation cost would help explain the underlying attraction of EMICs → Yes thanks, this has been lost from a previous version of the manuscript and will be added again, with the new text stating: "EMICs allow the integration of a large number of processes, more than conceptual models, using often coarser resolution and simplifying assumptions e.g. describing the atmospheric circulation compared to ESMs, but thus avoiding higher computational costs".

Pg. 3, ln. 75: the rationale for this omission of atmospheric deposition needs to be made very clear; I was previously assuming that you were ignoring it (1) because it was one more process to add (... but Landolfi et al. seem to have already added it to this model), and (2) because it's much less important than riverine input (... but you imply otherwise here); please be clear on the rationale here

As announced above, we add some explanations on our omission of atmospheric deposition in this paragraph starting l.68:

"Atmospheric deposition is known to be another important source of N to the ocean. Although it is estimated to add nitrogen at the same magnitude as the rivers and will also become more important with increasing anthropogenic activities (e.g. Tyrell, 1999; Cornell et al., 1995), it will not be considered in this study. Previously, Landolfi et al. (2017) and Somes et al. (2016) used UVic to study the response of the marine N cycle to atmospheric N deposition and its impact on marine productivity. While Somes et al. (2016) performed a series of idealized sensitivity experiments to evaluate the spatial and temporal scales of N cycle feedbacks, Landolfi et al. (2017) used an atmospheric N deposition forcing reconstructed using the multimodel mean of the Atmospheric Chemistry and Climate Model Intercomparison Project (Lamarque et al., 2013). Both found that N cycle feedbacks stabilize

the model's marine N inventory and limit changes to the marine N cycle and productivity. But none of these studies included riverine N supply. In order to disentangle the effects of the different sources of N, we are using UVic without atmospheric N deposition, to focus on…"

Pg. 3, ln. 79: "series of simulations" - it would possibly be helpful if these experiments were given a scientific rationale in addition to their description (e.g. "this experiment simulates increased anthropogenic emissions", "this experiment simulates differential waste water management between geographical regions", etc.)

→ We have modified the text as follows:

""

→ "In order to test the N cycle mechanisms and feedbacks found and described before, we set up an experiment where we simulate differential riverine nitrogen supply to the coastal oceans."

Pg. 4, Figure 1: this diagram makes it look like PO4 might be added to the ocean as well via a fixed R_N:P parameter; is that right?

→ NO3 and PO4 are linked through exchanges with the biological variables by constant (Redfield) stoichiometry, but PO4 is not directly added while NO3 is supplied from the rivers. This has been added in the caption of Fig.1 (see below).

Pg. 4, Fig. 1: maybe identify the 7 state variables in the caption to help with clarity

→ The caption of Figure 1 has been updated to be: "Ecosystem model schematics for the NPZD model with the prognostic variables (in square boxes) and the fluxes of material between them, indicated by arrows. The prognostic variables include two nutrients, nitrate ($NO_3$) and phosphate ($PO_4$), two phytoplankton (nitrogen fixers $P_D$ and other phytoplankton $P_O$) as well as zooplankton (Z), sinking detritus (D), and dissolved oxygen ($O_2$). Nitrate ($NO_3$) and phosphate ($PO_4$) are linked through exchanges with the biological variables by constant (Redfield) stoichiometry."

Pg. 4, ln. 97: "updates" - a little expansion on these updates might help readers understand if they are significant

→ We included: "… with updates of some of the equation parameters as noted in Partanen et al. (2016), where a small error in the code corrected".

Pg. 4, ln. 97: "prognostic variables" - you don't refer to the variables by the abbreviations used in Figure 1; nor does Figure 1's caption

→ Thank you for this reminder. The caption has been updated (see above) and the lacking abbreviations will be included in the text: "Seven prognostic variables are embedded within the ocean circulation: two phytoplankton classes (nitrogen fixing diazotrophs $P_D$ and other phytoplankton $P_O$), zooplankton (Z), sinking particulate detritus (D), nitrate ($NO_3$), phosphate ($PO_4$) and oxygen ($O_2$). Nitrate ($NO_3$) and phosphate ($PO_4$) are linked through exchanges with the biological variables by constant (Redfield) stoichiometry. "

Pg. 4, ln. 100: "atmospheric" - technically, they're fixing dissolved dinitrogen which is in equilibrium with atmospheric N2
→ The text has been changed to: "Nitrogen gas dissolved in seawater "

Pg. 5, ln. 105: "benthic denitrification" - this could be described a little more clearly; I *think* you mean that a function based on a more sophisticated benthic sediment model turns receipt of organic matter at the seafloor into oxic and anoxic (with denitrification) remineralisation of this matter; is that right?
→ Thanks for your remark, we agree that our original text was confusing. We changed the text to: "Benthic denitrification, in particular, is believed to be the major sink for fixed N (Voss et al., 2013; Galloway et al., 2004). It is included here through empirical transfer functions derived from benthic flux measurements (Bohlen et al., 2012). The functions are based on the rain rate of particulate organic carbon to the seafloor and bottom water $O_2$ and $NO_3$ concentrations."

Pg. 5, ln. 107: "models" - "models" or "model"?; this is a little confusing → see above

Pg. 5, ln. 109: "subgrid" - how "subgrid" is subgrid here?; for instance, do you simply identify the fraction of a cell that is shelf and do calculations based on this basic split, or do you divide each cell into an N x M subcell domain that does the bathymetry better?
→ This information has been added: "For each cell near the coast this scheme calculates the sea floor area with the cell at a higher resolution following Somes et al. (2010b)".

Pg. 5, ln. 123: To avoid confusion in readers, perhaps mention that NEWS2 includes no runoff from the Antarctic continent → We will add to the section the sentence: "Note that NEWS2 excludes runoff from the Antarctic continent."

Pg. 5, ln. 133-135: it sounds like this assumes a constant seasonal cycle in runoff; this is not unreasonable in an EMIC where the hydrological cycle may be in a long-term equilibrium → Yes, we can add this comment in the text: "Here, we assumed a constant seasonal cycle in runoff and that nitrogen concentrations in the discharged river water are constant throughout the seasonal cycle. We then distributed the annual load over the months, weighted by the fraction of monthly freshwater discharge."

Pg. 6, ln. 136: can you clarify what happens with riverine P in this model please?; Figure 1 tends to imply there might be a link between riverine N and P; Global NEWS provides both, and the balance of N and P could be important for your model's N2-fixation response
→ "Phosphate is not added in this experiment and we generally assume a fixed marine P inventory, like in most previous studies with UVic. The inclusion of a dynamic P cycle (like in Niemeyer et al., 2017, Kemena et al., 2019) with riverine P supply from NEWS2 will be subject to a follow up study".

Pg. 6, ln. 145: "bioavailable" - maybe add: "(or rapidly turned over to DIN)" → Yes, we can add that
Pg. 7, Table 1: add control experiment to this table → Thank you for this suggestion, CTR will be added
Pg. 7, ln. 149: "steady state" - it might be nice to see a figure (supplementary?) where the long-term balancing of the N budget took place; for instance, to illustrate the

timescales associated with addition and removal processes → Thank you for this suggestion. We had included the timeseries of N in a previous version and planned to prepare it for the supplements. We will submit it with the revised manuscript.

Pg. 7, ln. 155: "fairly well" - all runs omit the midwater maximum around 1000 m → "In comparison with observational data of the World Ocean Atlas (Garcia et al., 2019), the model underestimates the observed $NO_3$ in the whole water column by 3 to 4 mmol m$^{-3}$ and especially omits the midwater maximum around 1000 m."

Pg. 7, ln. 156: "global average" - what is the global concentration difference?; and what is this as a percentage of total observed inventory?
→ The text will be updated to read: "Global average NO3 concentrations only vary a little between the simulations (from 22.19 mmol m-3 in CTR to 22.48 mmol m-3 in NEWS and 22.84 mmol m-3 in 2xDIN) and the differences to CTR correspond to +1.1 %, +1.8 % and +2.5 % of the total observed inventory for NEWS, DIN+DON and 2xDIN respectively. However, the absolute error between model and observations decreases with higher riverine N supply."

In addition to the numbers in the text and in Table 2 we will include some of these numbers here. Note that the differences relative to the observed inventory correspond to the last column in Table 2.

| Simulation | Glob. NO3 Conc. [mmol m-3] | NO3 surface | NO3 1100 m | Difference to CTR | Diff / WOA [%] |
|---|---|---|---|---|---|
| CTR | 22.19 | 5.71 | 24.64 | 0 | |
| NEWS | 22.48 | 5.93 | 24.94 | 0.288 | 1.1 |
| DIN+DON | 22.66 | 6.15 | 25.10 | 0.468 | 1.8 |
| HIGH DIN | 22.84 | 6.19 | 25.30 | 0.646 | 2.5 |
| WOA | 26.29 | 6.62 | 31.22 | 4.102 | |

Pg. 7, ln. 156: "misfit" - this is absolute error, right? → The text has been changed accordingly (see above)
Pg. 7, ln. 167: "not surprising" - well, not *entirely* surprising; it's possible, of course, that adding local sources of N might trigger strong balancing denitrification that could even offset the addition (though this seems unlikely) → Yes, we will attenuate this statement as you suggest.
Pg. 7, ln. 171: on this point, might it be possible to include some total of suboxic ocean volume (e.g. the volume of below some standard oxygen threshold concentration)?
→ We have calculated global volume of oxygen minimum zone (OMZ) for all simulations, so that we can include a sentence here: "Note, that global volume of ocean minimum zones, defined here as regions with $O_2$ concentration lower than 70 µmol kg$^{-1}$, are increasing with higher nitrogen supply, from 52 *10$^6$ km$^3$ in CTR, to 54 *10$^6$ km$^3$ in NEWS, 56 *10$^6$ km$^3$ in DIN+DON and 58 *10$^6$ km$^3$ in 2xDIN."

Pg. 8, Table 2: "+1,12" -> "+1.12" → The typo will be corrected.
Pg. 10, Figure 5: normally, delta (or bias) plots are coloured blue (negative) to red (positive) with white in the centre –> Thank you for this suggestion, we have replotted all delta figures with a red-white-blue colourbar.

Pg. 11, Figure 6: why 850m?; from Figure 3b, the largest misfit seems to be at 1000m → Figure 6 has been updated with the blue to red colourbar. We chose 850 m and 1100 m depth, because although in the global profiles the largest misfit has been found around 1000 m, at 850 m depth the regions with decrease in No3 concentrations are most pronounced (compare also with Figure 5).

Pg. 12, Figure 7: it's obviously not possible to tell how realistic oxygen is here; perhaps compare NEWS and CTR directly to WOA instead? → If we would compare NEWS and CTR to WOA, we would find almost no differences between the two plots, because our UVic simulations show a somewhat different pattern and magnitude in $O_2$ especially in the Arabian Sea and the Gulf of Bengal. What we have done to address your comment is to add four plots to the Figure, showing the differences between WOA and CTR for $O_2$ and $NO_3$ in the two ocean basins.

Pg. 13, ln. 192: surplus "like" → Yes, corrected, thank you.

Pg. 13, ln. 193: assuming that N inputs fuel corresponding increases in productivity, it may be worth noting in passing how much production is also enhanced by river N in these regions → We haven't calculated total production in the Bay of Bengal but the analog Figure to Figure 9 and 10 for NPP shows, that productivity is enhanced only very locally near the coast of Myanmar (but up to +36 Tg C yr-1). In the rest of the basins, productivity is lower in the simulations with riverine N supply. We will include this Figure in the supplements and add the information in the text.

Pg. 13, ln. 197: see my earlier remarks about PO4 availability and riverine sources; if riverine P is neglected, this may skews where N2 fixation is favoured; ditto if PO4 is added in strict proportion to DIN → P is not altered by the additional nitrogen from river supply

Pg. 14, Figure 8: maybe it would be better to show CTR N2 fixation in addition to the deltas? → An additional plot has been added to the panel showing $N_2$-fixation from CTR simulation.

Pg. 15, ln. 223: "oxygen concentrations even though higher at the surface" - Is this elevated oxygen due to enhanced production?; typically surface oxygen is boring because it equilibrates quickly to saturation values (with ambient temperature)
→ We have added here: "In the Bay of Bengal, oxygen concentrations appear higher at the surface in NEWS than in CTR by around 1.5 mmol m$^{-3}$ at least in the southern part of the Bay, which could be due to enhanced production. However, compared to WOA, the oxygen concentrations are still very low in the upper 800 m in NEWS like in CTR, and they are particularly low in the NEWS simulations in the subsurface waters and the whole deeper basin (Fig. 7)."

Pg. 15: per my remarks for Figure 7, making the model's relationship with observed oxygen clear might be useful → Thanks for this suggestion, a comparison with $O_2$ from WOA has been added to the panel plot.

Pg. 17, Table 3: add a column listing the balance at equilibrium?; I make the discrepancy

about 0.5 Tg N / y for all of the model experiments; actually, what is this discrepancy?; is it the model just not fully equilibrated? → Yes, there are discrepancies between 0.4 and 0.5 Tg N / yr in the total sums depending on the simulation, although the model is equilibrated. Nevertheless, UVic has some natural variability and N fixation and denitrification oscillate with a difference in the range of the discrepancies you mention. We will add this information after the table: "Note that the global sums from sources and sinks do not exactly add to zero due to natural variability in the modeled N-cycle."

Pg. 18, Figure 11: if possible, it might be an idea to include a map of observational estimated production; the total of the models here might be OK, but I think the patterns - particularly in the Indian Ocean - might not be; this is important given the amount of analysis that is focused in this region → We will provide a Figure with marine production estimated from satellite observations, based on data from Behrenfeld and Falkowski 1997.

Pg. 18, Figure 11: surplus colourbar on Figure 11b? → Figure 11 has been updated with a new (blue-red) colour scheme and without the colourbar on panel (b).

Pg. 19, Table 4: this is a weird, single-column table → We will include more lines. The table will now appear as:

| Source | NPP [Pg C yr-1] | Method |
|--------|-----------------|--------|
| UVic CTR | 54.9 | model |
| UVic NEWS | 55.3 | model |
| UVic DIN+DON | 55.5 | model |
| UVic 2xDIN | 55.7 | model |
| Behrenfeld and Falkowski (1997) | 43.5 | Satellite data |
| Behrenfeld et al. (2005) | 67 | Satellite data |
| Carr et al. (2006) | 51 | Mean of 31 global models |
| Westberry et al. (2008) | 52 | Carbon based, spectral |
| Buitenhuis et al. (2013) | 56 | Model and observationsal database |

Pg. 19, Table 4: these numbers are a bit higher compared to what I'm used to; e.g. the Oregon State University primary production website; there, I find ~40 Pg C / y; the 3 models their site includes are quite divergent, however
→ Buitenhuis et al (2013) shows an overview from 14 different approaches to calculate global marine primary production rates and the recent one's range between 40 and 60 Pg C / y. From the studies cited by Buitenhuis, the global production as calculated by Westberry et al. (2008) and Behrenfeld et al. (2005) are also in Table 4. We have also analysed the dataset from Behrenfeld and Falkowski (1997), which gives a rate of 43.5 Pg C yr$^{-1}$. To be more complete, we will also include this study in Table 4 (see answer below).

Pg. 20, Table 5: this is a very strange way to organise a table; it's a single column when it should be three columns of numbers→ Yes, sorry for this. Originally, we hadn't planned to provide this table and included it at the last moment. It will be reorganized in the revised manuscript as you suggest, including supplementary numbers (see responses below).

Pg. 20, Table 5: also, I might be inclined to include numbers (or deltas) for the other major N-cycle processes, N2-fixation and denitrification → Table 5 was originally only a supplement for Figure 11. We haven't extracted all the minima and maxima for the other N-cycle processes but for the revised manuscript, we have calculated the global totals for the regions presented in this table and will include them as suggested.

Pg. 20, Table 5: also, why the areal units here?; would it not make more sense to report global totals (i.e. Pg C / y)? → Table 5 was originally only a supplement for Figure 11, to show the extremes in vertically integrated primary production rates. For the revised manuscript we will include the regional totals in Tg C / yr for the Bay of Bengal, the Yellow Sea, the North Sea, the outflow of the Rio de la Plata and the East Mediterranean Sea to put the numbers into relation. This will then look like this (additionally we will include N-cycle processes):

| Region | Difference in NPP in the whole region [Tg C yr-1] | | |
|---|---|---|---|
| | NEWS - CTR | DIN - CTR | 2xDIN - CTR |
| Bay of Bengal | -19 | -23 | -31 |
| Yellow Sea | 119 | 133 | 190 |
| North Sea | 35 | 43 | 62 |
| Rio de la Plata river mouth | 17 | 27 | 35 |
| Eastern Mediterranean Sea | 41 | 50 | 53 |

Note that local minima or maxima can be considerably higher than the global sums.

Pg. 24, ln. 352: "increase in marine primary production is small" - this analysis appears not to factor in that total riverine input of N is ~0.2% of the N used in primary production; so, contrary to the point you make here, the changes in NPP found between the simulations appear to actually be quite large; I guess the factor that makes the rivers more important is that they deliver N to shallow ocean areas (= shelves) where they will have a larger impact
→ Yes, thanks for your remark. Unfortunately, I could not find a reference for this number (0.2 %). But we will add to our text: "The absolute increase in marine primary production is small (between +0.7 % in NEWS and +1.3 % in 2xDIN). However, relative to the amount of N added to the global ocean, primary production increases yearly by 17.5 Tg C per additional Tg N in NEWS (16.0 Tg C per Tg N in 2xDIN). As we have shown, primary production increases mainly near the river mouths, where high nutrient loads are injected in shallow ocean areas, creating production "hot spots", while only small changes in production have been found in the open ocean. Other studies with additional N supply also found only moderate increase in global primary production rates…"

Pg. 25, ln. 387: "carbon export" - how is this defined?; e.g. 100 m export?; and does it include or exclude shelf regions where material is not properly exported? → We will add in the text: "Indeed, simulated carbon export, evaluated at the 82.5 m level and including all the shelf regions, increases globally by only 0.06 Pg C / yr in our NEWS simulation."

Pg. 25, ln. 387: You might want to consider these changes in light of how much nutrient is being added to the ocean relative to N cycling through production; if my earlier back-oftheenvelope calculations are right, river N is 0.2% of N-cycling through production, but you're finding several percent in NPP change

→ We modify our conclusion to: "We have seen that in the coastal regions and especially in some hot-spot regions near the river mouths, riverine nitrogen input leads to higher primary production rates. Globally, NPP rates increase up to 17.5 Tg C per Tg N added to the ocean. However, the biogeochemical feedbacks of the ocean buffer further increases in global N concentrations and global NPP."

Pg. 25, ln. 390-392: this paragraph doesn't really say very much; I'd suggest deleting it
→ We will shorten it to: " We have found, that likewise to atmospheric deposition, river supply of nitrogen is not only relevant for the coastal system but also for marine biology in the global ocean. But while atmospheric deposition provides only N, …"

Pg. 26, ln. 397: if riverine phosphorus is not included here, does that not skew the model's balance between N2-fixation and denitrification since the N being added is not balanced by P? → Yes, this may be true, which is why we will investigate P and N additions in a following study

Pg. 26, ln. 400: "hdl" - what is this?; could this be put onto Zenodo or something to get a proper DOI? → The correct link was https://hdl.handle.net/20.500.12085/59977a36-e8e7-4348-a4e8-2b13f3913590 and will be corrected in the revised manuscript. We are still waiting for the OPeNDAP access. Thank you for the suggestion of Zenodo. I have planned to use this server for the next data set.

Pg. 26, ln. 402: I get a 404 error from this link → I tested the link again and it worked for me. Perhaps we could just include the "Home" link which is http://icr.ioc-unesco.org/? In the index on the left it is also possible to find the link to the "Global NEWS".

**Style points:**
Pg. 8, Figure 3: can you make the lines in the key thicker so it's easier to tell them apart?; also, the choice of colours is rather unhelpful in this regard; also, why are they ordered in this strange way in the key? → Thanks for your suggestions. We have changed the lines and the colours of the Figure.
Pg. 9, Figure 4: this colour scale looks more like a delta one to me → We have changed the colours of the Figure.

Pg. 10, Figure 5: it might be better not to stretch the smaller Indian basin to the same size as the other basins here; perhaps just plot the same latitude range on all three panels? → As this would mean a lot of work to change the range of all Figures with the ocean basins, we have let the plot of the Indian Ocean like it is. But we will include a note, indicating that the ranges are different.

Pg. 15, Figure 9: this colourbar implies negative denitrification is possible in the model; is

it? → We have changed the colourbars of this Figure, using a delta colourbar (blue-red) for panels b-d and have rearranged the colourbar for panel (a) in order to avoid this impression. There are no negative denitrification rates in (a).

Pg. 16, Figure 10: this colourbar is missing the extreme cyan colour that indicates "out of range" delta concentration → Thanks for you remark. The colourbar has been changed to blue-red and the cyan colour is now shown correctly.

Pg. 23, Figure 13: could you try a clever log scale here?; this plot is otherwise not very Informative → Unfortunately we haven't found another way to plot these results.

Pg. 25, ln. 369: purely as a style point, I would suggest thinning your conclusions section to 5 or 6 bullet-point statements of your findings; this makes it very easy for readers to understand the main findings (and decide whether to read more!) → Even though I am not very fond of bullet points in a text, I will follow your suggestion for the revised manuscript.

**Additional literature to the manuscript**

Behrenfeld M. J. and Falkowski P. G.: Photosynthesis rates derived from satellite-based chlorophyll concentration, Limnology and Oceanography, 42, 1-20, 1997.

Capone D. G., Zehr J. P., Paerl H. W., Bergman B., and Carpenter E. J.: Trichodesmium, a Globally Significant Marine Cyanobacterium, Science, 276, 1221-1229, 23 May 1997. Cornell S., Rendell A., and Jickells T.: Atmospheric inputs of dissolved organic nitrogen to the oceans, Nature, 376, 243-246, 20 July 1995.

Kim I.-N, Lee K., Gruber N., Karl D. M., Bullister J. L., Yang S., and Kim T.-W.: Increasing anthropogenic nitrogen in the North Pacific Ocean, Science, 346, Issue 6213, 1102-1105, 28 November 2014.

Krishnamurthy A., Moore J. K., Zender C. S., and Luo C.: Effects of atmospheric inorganic nitrogen deposition on ocean biogeochemistry, J. Geophys. Res., 112, G02019, doi:10.1029/2006JG000334, 2007.

Lamarque J. F., Dentener F., McConell J., Ro C.-U., Shaw M. et al.: Multi-model mean nitrogen and sulfur deposition from the Atmospheric Chemistry and Climate Model Intercomarison Project (ACCMIP): evaluation of historical and projected future changes, Atmos. Chem. Phys., 13, 7997–8018, 2013.

Oschlies A., Koeve W., Landolfi A., and Kähler P., Loss of fixed nitrogen causes net oxygen gain in a warmer future ocean, nature communications, 10:2805, doi.org/10.1038/s41467-019-10813-w, 2019.

---

## Author Response (AR1)

**Comment on bg-2021-101**
**Anonymous Referee #1**

Referee comment on "Riverine nitrogen supply to the global ocean and its limited impact on global marine primary production: a feedback study using an Earth System Model" by Miriam Tivig et al., Biogeosciences Discuss., https://doi.org/10.5194/bg-2021-101-RC1, 2021

This is an interesting article exploring the long-term impact of riverine nitrogen inputs on the global ocean nitrogen inventory and associated primary production. The authors show that in simulations that have reached equilibrium, impacts on the global N inventory and primary production are highly limited due to feedbacks on N fixation and denitrification. I appreciated the candidness of the discussion on the modelling approach's strengths/weaknesses and how the results differ from those previously published in this area. The main discussion that I think might be expanded upon is how indicative simulations that have reached equilibrium are for policy relevant timescales (interannual to decadal). Put another way, how do the authors' main conclusions change over the course of these 10000-year simulations? Are the implications different for watersheds that are experiencing rapid increases or decreases in nitrogen export at present? Most of my comments and suggestions relate to how the manuscript text could be better structured and the figures could be made much easier to interpret. Subject to these changes, I would be happy to recommend for publication.

→ Thank you for your recommendation and positive feedback.
Regarding your question, how the main conclusions change over the whole simulation, we have included additional text in the revised manuscript, indicating that the main changes in the global marine nitrogen inventory appear in the first 4000 years of the simulation. After this time, the global nitrogen budget is almost in equilibrium. In the supplements we submit a Figure showing the timeseries of yearly N fluxes during the 10 000 years of the simulation After the perturbation by additional nitrogen input via the rivers, marine primary production increases globally in all our simulations, but only for less than 2000 years.
Interannual to decadal timescales are shorter than our current model time resolution. The model can be used on annual to decadal timescales, but not on global (spatial) scales. While globally the feedbacks take thousands of years to balance the N budget, locally the feedbacks may be much faster. For a next study it can be an interesting question to analyse N2 fixation, denitrification and the vicious cycle on shorter timescales in regions experiencing rapid increases (or decreases) in nitrogen export due e.g. to anthropogenic activities, like in the East China Sea.

**Minor comments**

L23. "atmospheric" –should probably be dissolved/aqueous.
We have expanded the sentence to "Biological dinitrogen (N2) fixation refers here to atmospheric $N_2$, dissolved in seawater, which is reduced to ammonia." (L24)

L26. Aren't these "model concepts" observationally /experimentally derived? Some reference to the empirical evidence would be useful here. Maybe also its limitations if it's heavily based on given species (eg Trichodesmium).

We appreciate and have followed the reviewer's comment by referring to earlier work on empirical evidence and how it is used in conceptual and numerical models:
- Karl et al. (2002): Dinitrogen fixation in the world's oceans, Biogeochemistry 57/58: 47–98.
- Landolfi et al. (2018) Global Marine N2 Fixation Estimates: From Observations to Models.Front. Microbiol. 9:2112. doi: 10.3389/fmicb.2018.02112
- Zehr and Capone (2020): Changing perspectives in marine nitrogen fixation, Science 15 May 2020, Vol. 368, Issue 6492, eaay9514, DOI: 10.1126/science.aay9514

We now also indicate that a part of our knowledge of N2-fixation is still based on the original limited assumptions based on given species especially Trichodesmium. (L31-33)

L31. concentrations "of" fixed N → corrected (L.40)

L32. "the consumption of O2"- Do you mean the consumption of O2 during remineralisation? If so, be explicit. → "heterotrophic O2 consumption during organic matter remineralization" (L40)

L47. The Séférian et al. 2020 reference is perhaps worth citing here as it summarises the inclusion of riverine inputs in recent models. → is now included (L63)

L51. "real" should probably be realistic or observed. → "without observed nutrient fluxes" (L67)

L59. Maybe clarify what is meant by N export here. Riverine delivery? For many in the ocean biogeochemistry community export is instinctively a vertical flux. → "Their focus was on pre-industrial nutrient input from the rivers" (L75)

Figure 1. More detail is needed on the tracers in the figure legend. → We have changed the caption to "Ecosystem model schematics for the NPZD model with the prognostic variables (in square boxes) and the fluxes of material between them, indicated by arrows. The prognostic variables include two nutrients, nitrate ($NO_3$) and phosphate ($PO_4$), two phytoplankton (nitrogen fixers $P_D$ and other phytoplankton $P_O$) as well as zooplankton (Z), sinking detritus (D), and dissolved oxygen ($O_2$). nitrate ($NO_3$) and phosphate ($PO_4$) are linked through exchanges with the biological variables by constant (Redfield) stoichiometry."(P5, Figure1)

L100. "atmospheric" –should probably be dissolved/aqueous. → "Nitrogen gas dissolved in seawater "(L127)

L101. Are they limited by a max NO3 concentration? I know much of this will be in the cited references but more detail is required on N-fixation in the model. Highlight perhaps that most models don't have explicit diazatrophs and this is an advantage of using uvic. What is the diazatroph PFT based on? How does N-fixation compare to observations where they exist?

→ We have included a more detailed description in the revised manuscript: "Since diazotrophs can fix nitrogen gas dissolved in seawater, they are not limited by NO3 nor by a maximum NO3 concentration, while the growth of other phytoplankton is limited by NO3 and PO4 (note that both are additionally limited by iron, light and temperature). The explicit integration of diazotrophs permitting the computation of nitrogen fixation, is not given in all ocean models but makes UVic a good choice to study nitrogen cycle feedbacks. The

maximum potential growth rate of diazotrophs is not only based on temperature as in most models, but also on dissolved iron, which is necessary e.g. for photosynthesis or the reduction of nitrate to ammonium (Keller et al., 2012; Galbraith et al., 2010). Keller et al. (2012) found that the observational estimates were within the range of global nitrate fixation rates from estimations and the patterns of N2 fixation from the new model were mostly consistent with the relatively sparse available observations (Sohm et al., 2011).)." (L.127-135)

L105-110. As with the above comment, some comparison on how denitrification in the model compares to observations would be very useful. I think a global map of N-fixation and denitrification in the model CTR is required.
→ In the revised manuscript we include maps showing the global distribution of $N_2$-fixation and denitrification (Figure 8a and Figure 9). A more detailed discussion of spatial patterns is added to the text. (L263-271)

L140. It should be made clearer on first use that NEWS etc are simulation names. → We will include a sentence about the simulation names at this place. (L173)

Table 1. For clarity I would remove UVic from the simulation names as this is not repeated in the main text. I would also add a CTR row. → Yes, absolutely right. This has been done throughout the revised manuscript. (P.8)

L156. "vary a little" – please quantify this → "Global average NO3 concentrations only vary a little between the simulations (from 22.19 mmol m-3 in CTR to 22.48 mmol m-3 in NEWS and 22.84 mmol m-3 in 2xDIN) (L.196)

L151. This should really be called a "Results and Discussion" section. → Yes, some of the final discussions have been directly included in the results chapter, so we have renamed this section as you suggest. (L. 191)

L160. The wording here needs to be clearer. "At smaller scales...globally higher." This reads like it contradicts L167-168. → In the revised manuscript we change this sentence to "Nevertheless, in all three simulations (NEWS, DIN+DON, and 2xDIN), NO3 concentrations are globally higher compared to the control simulation (CTR)..." (L199)

Figure 3. Axes are missing labels and units here. → This has been corrected in the revised Figure 3 (P9)

Figure 5. I find it very difficult to see differences between positive and negative anomalies using this colorbar. I suggest changing to something far more distinctive (e.g. red for negative anomalies). The same applies to other figures using this scale. → All Figures concerned have been updated with a new colorbar (in blue and red). We have chosen a delta colorbar with blue for negative and red for positive values.

L175-177. This is difficult to see in Figure 5 maybe cite figure 6 here. → Citation of Figure 6 included here. (L200)

Figure 6. The depths given in the figure don't match the legend. → Figure 6 has been updated for the revised manuscript. (Figure 6, P12)

L181-183. This sentence is confusing and needs rephrasing.
→ We have improved the text to read: "But in the deeper northern Indian Ocean basin down to approximately 2000 m, NO3 concentrations are significantly lower in NEWS than in CTR. Considering the zonal average of the Indian Ocean, NO3 concentrations are lower by -0.7 to -0.9 mmol N m$^{-3}$, and even more if only the zonal average of the Bay of Bengal is considered." (L222)

Figure 7. Label missing from panel c. → Figure 7 has been updated for the revised manuscript. (P14)

L187. typo. "amounts to an increase of only 1.1…" → corrected (L229)

L209. I'm not sure the language here is accurate. Presumably the model is not explicitly trying to compensate anything. Wouldn't this be better described as enhanced denitrification sinks promoting conditions that favour N-fixers over the other PFT type and consequently global N-fixation rates are higher?
"*To compensate for this additional N sink, the model estimates higher fixation rates.*" changed to:
→ "The additional N sink in form of benthic denitrification promotes conditions that favor N-fixers, i.e., diazotrophs, leading to higher nitrogen fixation rates." (L261)

L210-213. The discussion of other literature here before properly explaining your model results is confusing. Where these papers have used the same model this should be clear.
→ Thanks for pointing this out. The paragraph has been reorganized, see comment below about L219-227.

L216. "where NO3 concentrations are substantially reduced relative to the CTR" → The text has been updated, thanks for the rephrasing.

Figure 8- See earlier comments on how it would be nice to see global Nfix in the control.
→ Thanks for the suggestion, this has been done in the revised Figure. (Figure 8, P16)

Figure 9. Suggest using different color palettes for mean states and anomalies. → Thanks for the suggestion, this has been done in the revised Figure (see also response to Figure 5)

L219-227. The balance between results of the model simulations and the discussion of other literature needs to be more organised. The presentation of discussion before results is quite confusing.
→ To address this critique, the whole paragraph (previously l. 190-241) has been restructured and divided in two subsections: "3.1.2. Denitrification and nitrogen fixation" and "3.1.3 The N-cycle feedback mechanisms"). (L232-318)

L221-222. This is a bit rushed and therefore confusing. I think more detail is needed here on this mechanism, the difference between the stoichiometry of N-fixation and denitrification and how spatial and temporal coupling is important for the positive

feedback to occur.

→ Sorry that this was confusing. We have now updated the text to read, "This is due to the stoichiometric imbalance created by the combination of these processes. Denitrification occurs in anoxic or suboxic environments, where nitrate or nitrite can be used as a substitute terminal electron acceptor instead of oxygen. Denitrification consumes 7 mol of NO3 for every mole of organic N provided by N2-fixation and remineralized anaerobically via denitrification. If more than 1/7 of the organic N provided via N2 fixation is denitrified, this leads to a net loss of N by more N lost during denitrification than added via N2 fixation, called 'vicious cycle' by Landolfi et al. (2013)." (L295-300)

Table 3. Benthic denitrification appears to be twice the magnitude of that in the water column. This doesn't seem to be reported and discussed in the manuscript. Does this have implications for models lacking benthic denitrification?

→ We don't see, where benthic denitrification appears to be twice the amount of water column denitrification (WCD), but indeed benthic denitrifications (BD) is an important process (e.g. Somes et al., 2016; Somes et al., 2013) and for both global estimates vary considerable: for WCD estimates are between 50 and 150 Tg N yr-1, for BD between 100 and 300 Tg N yr-1 (Galloway et al., 2004; Gruber, 2004; Bohlen et al., 2012; Somes et al., 2013). In the revised manuscript we address this point. Our global results for N2 fixation, WCD and BD stay in the range assumed for a balanced fixed-N budget in the preindustrial ocean e.g. by Somes et al. (2013). BD is more evenly distributed than WCD. Therefore, to study regional effects, it is helpful to include both processes. Nevertheless, models can have a balanced nitrogen budget without benthic denitrification. (Table 3, P18 and L239))

L246. "…and vary little between…" → corrected (L323)

L251. "smaller scales" is ambiguous. Here and elsewhere I recommend being more specific e.g basin/watershed/coastal scales etc.
→ Yes, at this place we changed the vague formulation to "Nevertheless, with rivers supplying N to the ocean, differences are visible at coastal scales: NPP increases locally, close to the river mouths (L327)

L257-259. Differences in spatial patterns between these simulations are difficult to discern it looks more like the magnitude of change is the only difference. → This impression is correct and we had added this remark to the text: "The main differences are in the magnitude of NPP, however, some regions with higher NPP can only be found in the simulation 2xDIN in the open ocean basins." Furthermore, we have changed the colorbar of the figure in blue and red like stated for Figure 5, in order to make the differences clearer. (L334)

L259. Are these subtropical and tropical regions where N-fixers are predominately confined to?
→ Yes, these are quite exactly the regions, where in UVic diazotrophs can be found. We have included a comment on this in the manuscript and refer to the new panel of figure 8. (L337)

L297. Maybe "exported again" should be "recycled" here. → yes. "recycled" is what was meant. (L381)

L301. And presumably not all the N can be consumed via local primary production due to other constraints.
→ Yes, we have included: "N is transported to the Indian basin but is not consumed by local primary production nor does it trigger the vicious cycle described before". (L 385)

Table 6. Maybe the increase in NPP per quantity of additional N would be a useful metric to add to this table given each watershed provides different total N delivery.
→ Yes, this is a good idea. This information has now been added to the existing table. (now Table 7, P25)

L325. For clarity I think "inhibiting additional NPP" should be "limiting increases in NPP".
→ Thanks for your suggestion, it has been changed. (L411)

Section 4. Given the extensive discussion of the Lacroix et al. 2020 paper, I think a few more details are required to compare the studies properly. What was their model resolution, did they have explicit N-fixers and benthic denitrification?
→ We have added the following information: "This is partly due to the coarser resolution of UVic. The grid configuration used by Lacroix et al. (2020) (GR15) consists of a bipolar grid which resolves the ocean horizontally at around 1.5° and through 40 unevenly spaced vertical layers. Although riverine N is simulated as fixed percentage of P, dynamic nitrogen fixation by cyanobacteria is included as well as nitrogen deposition and denitrification." (L416)

L358. The emphasis here and in the conclusions (L376) doesn't really match the findings. I would say the feedbacks do much more than "partly compensate" the riverine fluxes. Maybe this would be clearer if you gave the % of added N that is retained in the inventory at equilibrium or some other metric of feedback strength.
→ Yes, you are right. We have added: "Compared to the total amount of N added by the rivers at the end of the simulation, only 2.3 % (NEWS) to 2.6 % (2xDIN) is retained in the global inventory. The feedbacks compensate for much of the nitrogen addition and in some regions even overcompensate it."
The sentences in the conclusion were left unchanged, because some of the additional N is still accumulated in the deeper ocean, i.e. the net effect is a partial compensation. (L451)

L388-389. I'm not sure this "upper limit" conclusion would hold if N fertilisation were targeted spatially and temporally in regions of N limitation. Perhaps this should be toned down a little.
→ We agree that this needs a more detailed discussion and added the following text: "On short time scales target N fertilization might work, but once the vicious cycle has a chance to start, then the findings would probably be the same, as shown e.g. by Somes et al. (2016). But further research would need to be done for targeted spatial and temporal N additions at different levels in N limited regions." (L488)

**Response to the Referees comments – 02**

**Comment on bg-2021-101**
**Anonymous Referee #2**

**Major criticisms:**
- the model description has a few gaps that might benefit from filling and improved clarity
- the experimental design involves several highly idealised simulations and it could be clearer which hypotheses are being tested when these are being framed; for instance, rather than use a single experiment that scales river inputs, a suite of scaling runs could instead assess the strength and saturation of feedbacks
- the results are generally clear enough, but there are omissions or odd choices in the results presented; it would be helpful, for instance, to have tables which bring together the major N-cycle processes across the different runs

RESPONSE

Thank you for the comments and suggestions.
In the revised manuscript we address the major criticisms as followed:
- we have included more details in the model description and included the full description of the NPZD model as shown in Figure 1.
- In the revised manuscript we have added the hypotheses to the description of the experimental design. The first run, where we add the riverine DIN from the NEWS model is a base run to evaluate (i) how the model reacts to a realistic riverine nitrogen input and (ii) specifically how the nitrogen cycle responds to it. The following simulations, where we increased the amount of nitrogen added to the ocean, are already scaling experiments to test the mechanisms and feedbacks described for open-ocean nitrogen input via nitrogen fixation or atmospheric deposition by Landolfi et al. (2017) and Somes et al. (2016).
  We do not have the capacity to include new model runs in this study.
- Table 3 in the current manuscript brings together river supply in N, global N2 fixation, global benthic and global water column denitrification for the simulations CTR, NEWS, DIN+DON and 2xDIN.

**Minor comments:**

Pg. 1, ln. 1: I might be inclined to make a distinction between the natural and the anthropogenic nitrogen cycles; we have so radically modified the N-cycle that negative feedbacks may have been swamped; in any case, definitely: make it clear in the abstract whether you consider that you are dealing with the natural or modern N-cycle
→ Yes, that is correct. We mention it in the revised abstract (L1) and in the main text (L100), that we consider the natural N-cycle. Nevertheless, as stated in line 118, the NEWS dataset is based on river supply in the year 2000 and also includes natural and anthropogenic biogeophysical properties. See also response to next comment.

Pg. 1, ln. 18: "steady state" - I'm not enough of an expert in riverine supply to be sure,

but my first reaction is that the scale of anthropogenic inputs of nitrogen to the ocean must make this assumption questionable. It's an assumption I'm happy to make in my own tangential work for simplicity, but where a study is addressing it head-on, I'd expect something on the anthro perturbation to the N-cycle

→ We have addressed this very relevant remark in the introduction starting l. 54: "At this place it is appropriate to include some remarks about the anthropogenic perturbation of the N-cycle. In the Anthropocene, human activities have led to increased inputs of fixed nitrogen from the atmosphere and through different sources of runoff from land (e.g., Somes et al., 2016; Kim et al., 2014, Lamarque et al.,2013). At the same time, warming and deoxygenation can lead to increased N loss (Oschlies et al., 2019). The question has been therefore raised, if the global N budget could still be considered to be in steady state. While these combined effects on the N budget are still very uncertain, some studies suggest, that imbalances could be limited due to internal feedbacks of the N cycle (Landolfi et al., 2017; Somes et al., 2016; Krishnamurthy et al., 2006)." (L54)

Pg. 2, ln. 25: "slowly" - to assist less familiar readers, please expand on why we might expect (or why we know) diazotrophs to be slow-growing

→ New text has been added stating that: "Previous studies have shown, that at least the most commonly cultured diazotrophs, especially Trichodesmium, have low growth rates relative to many non-fixing phytoplankton (e.g. Capone et al., 1997). Indeed, while these organisms are able to fix $N_2$ when reactive nitrogen is scarce, this turns into an disadvantage in regions where N is more abundant, because $N_2$-fixation requires more energy (Tyrell, 1999). According to common conceptual models of controls on nitrogen fixation, the slowly growing diazotrophs are then out-competed by non-fixing phytoplankton, if enough P and other nutrients are present (Tyrrell, 1999). Note that part of our knowledge of $N_2$-fixation and most modern model concepts are still based on the original limited assumptions based on a few species, especially Trichodesmium. Also, emphasis has traditionally been put on bottom-up controls, despite accumulating evidence that top-down controls such as selective grazing on dominant species may have substantial impacts on the distribution of diazotrophs and nitrogen fixation (Landolfi et al., 2021)" (L26)

Pg. 2, ln. 29: "most marine organisms" - except diazotrophs, of course
→ "for most other marine organisms, except diazotrophs" (L37)
Pg. 2, ln. 32: "consumption of $O_2$" - this statement is perhaps confusing as people will be aware that growth of phytoplankton *produces* oxygen; I know what you mean here, but others might not
→ Thanks for your remark, we will change the text to: "heterotrophic $O_2$ consumption during organic matter remineralization" (L40)
Pg. 2, ln. 34: "work together" - do they really "work together"?; might it not be fairer to say that they work independently, but between them the nitrogen cycle is balanced (which it inevitably must be)
→ Text updated to read: "These two processes, $N_2$ fixation and denitrification, both contribute to regulating the global marine N budget." (L42)
Pg. 2, ln. 38: "considered in this study" - it might be useful to indicate the size of the deposition sink so that it's clear why it's ignored here; also, my first thought here was that you didn't want to have to add an additional low importance process, but it looks like someone has already done this for your model; so perhaps be very clear here why you're ignoring it

→ Thanks for your comment. We have addressed this in the revised version and deleted the sentence in l.38, but add some more explanations in l.85 (see comment to that line) " (L85)

Pg. 2, ln. 39: "highly" - how high is "highly"? do you have an estimate of how much the riverine flux has been affected by human activities, or is this still highly uncertain?
→ We have toned down this somewhat and include: "Although riverine N is not the main source of N for most marine environments, it can become important, as it is directly influenced by human activities. Seitzinger et al. (2010) e.g., estimated that global nitrogen export by rivers to the coastal waters increased by 17,7 % from 1970 to 2000. Nitrogen is known to impact…" (L47)

Pg. 2, ln. 47: "2008" - 2008 doesn't seem very "current"; perhaps reword or find a better Example → "However, often global biogeochemical ocean models still omit…" (L62)

Pg. 2, ln. 50: "as on" - "as on" -> "*than on* the actual quantity of nitrogen nutrient" → thank you, we have corrected this. (L67)

Pg. 2, ln. 55: "long enough" - you should say what you think the relevant time period is; I believe Tyrrell (1999) puts an estimate on this based on input flux and ocean inventory
→ "likely not long enough to study the feedbacks of the nitrogen cycle in the open ocean, considering that the mean residence time of fixed nitrogen in the ocean has been estimated to be a few thousand years (Gruber, 2004)." (L72)

Pg. 2, ln. 57: "estimations of" -> "estimated" → will be corrected
Pg. 3, ln. 62: "EMICs" - a passing mention of computation cost would help explain the underlying attraction of EMICs → Yes thanks, this has been lost from a previous version of the manuscript and will be added again, with the new text stating: "EMICs allow the integration of a large number of processes, more than conceptual models, using often coarser resolution and simplifying assumptions e.g., describing the atmospheric circulation compared to ESMs, but thus avoiding higher computational costs". (L82)

Pg. 3, ln. 75: the rationale for this omission of atmospheric deposition needs to be made very clear; I was previously assuming that you were ignoring it (1) because it was one more process to add (... but Landolfi et al. seem to have already added it to this model), and (2) because it's much less important than riverine input (... but you imply otherwise here); please be clear on the rationale here
We have added some explanations on our omission of atmospheric deposition in this paragraph starting l.85:
"Atmospheric deposition is known to be another important source of N to the ocean. Although it is estimated to add nitrogen at the same magnitude as the rivers and will also become more important with increasing anthropogenic activities (e.g., Tyrell, 1999; Cornell et al., 1995), it will not be considered in this study. Previously, Landolfi et al. (2017) and Somes et al. (2016) used the same model to study the response of the marine N cycle to atmospheric N deposition and its impact on marine productivity. While Somes et al. (2016) performed a series of idealized sensitivity experiments to evaluate the spatial and temporal scales of N cycle feedbacks, Landolfi et al. (2017) used an atmospheric N deposition forcing reconstructed using the multimodel mean of the Atmospheric Chemistry and Climate Model

Intercomparison Project (Lamarque et al., 2013). Both found that N cycle feedbacks stabilize the model's marine N inventory and limit changes to the marine N cycle and productivity. But none of these studies included riverine N supply. In order to disentangle the effects of the different sources of N, we are running the model without atmospheric N deposition, to focus on…"

Pg. 3, ln. 79: "series of simulations" - it would possibly be helpful if these experiments were given a scientific rationale in addition to their description (e.g. "this experiment simulates increased anthropogenic emissions", "this experiment simulates differential waste water management between geographical regions", etc.)
→ We have modified the text as follows:
""
→ "In order to test the N cycle mechanisms and feedbacks found and described before, we set up an experiment where we simulate differential riverine nitrogen supply to the coastal oceans." (L101)

Pg. 4, Figure 1: this diagram makes it look like PO4 might be added to the ocean as well via a fixed R_N:P parameter; is that right?
→ NO3 and PO4 are linked through exchanges with the biological variables by constant (Redfield) stoichiometry of organic matter, but no PO4 is added while NO3 is supplied from the rivers. This has been added in the caption of Fig.1 (see below). (Pg. 5)

Pg. 4, Fig. 1: maybe identify the 7 state variables in the caption to help with clarity
→ The caption of Figure 1 has been updated to: "Ecosystem model schematics for the NPZD model with the prognostic variables (in square boxes) and the fluxes of material between them, indicated by arrows. The prognostic variables include two nutrients, nitrate ($NO_3$) and phosphate ($PO_4$), two phytoplankton (nitrogen fixers $P_D$ and other phytoplankton $P_O$) as well as zooplankton (Z), sinking detritus (D), and dissolved oxygen ($O_2$). Nitrate ($NO_3$) and phosphate ($PO_4$) are linked through exchanges with the biological variables by constant (Redfield) stoichiometry of organic matter." (Pg. 5)

Pg. 4, ln. 97: "updates" - a little expansion on these updates might help readers understand if they are significant
→ We included: "… with updates of some of the equation parameters as noted in Partanen et al. (2016), where a small error in the code corrected". (L122)

Pg. 4, ln. 97: "prognostic variables" - you don't refer to the variables by the abbreviations used in Figure 1; nor does Figure 1's caption
→ Thank you for this reminder. The caption has been updated (see above) and the lacking abbreviations will be included in the text: "Seven prognostic variables are embedded within the ocean circulation: two phytoplankton classes (nitrogen fixing diazotrophs $P_D$ and other phytoplankton $P_O$), zooplankton (Z), sinking particulate detritus (D), nitrate ($NO_3$), phosphate ($PO_4$) and oxygen ($O_2$). Nitrate ($NO_3$) and phosphate ($PO_4$) are linked through exchanges with the biological variables by constant (Redfield) stoichiometry of organic matter. " (L124)

Pg. 4, ln. 100: "atmospheric" - technically, they're fixing dissolved dinitrogen which is in equilibrium with atmospheric N2
→ The text has been changed to: "Nitrogen gas dissolved in seawater "(L127)

Pg. 5, ln. 105: "benthic denitrification" - this could be described a little more clearly; I *think* you mean that a function based on a more sophisticated benthic sediment model turns receipt of organic matter at the seafloor into oxic and anoxic (with denitrification) remineralisation of this matter; is that right?
→ Thanks for your remark, we agree that our original text was confusing. We changed the text to: "Benthic denitrification, in particular, is believed to be the major sink for fixed N (Voss et al., 2013; Galloway et al., 2004). It is included here through empirical transfer functions derived from benthic flux measurements (Bohlen et al., 2012). The functions are based on the rain rate of particulate organic carbon to the seafloor and bottom water $O_2$ and $NO_3$ concentrations." (L137)

Pg. 5, ln. 107: "models" - "models" or "model"?; this is a little confusing → see above

Pg. 5, ln. 109: "subgrid" - how "subgrid" is subgrid here?; for instance, do you simply identify the fraction of a cell that is shelf and do calculations based on this basic split, or do you divide each cell into an N x M subcell domain that does the bathymetry better?
→ This information has been added: "For each cell near the coast this scheme calculates the sea floor area with the cell at a higher resolution following Somes et al. (2010b)". (L144)

Pg. 5, ln. 123: To avoid confusion in readers, perhaps mention that NEWS2 includes no runoff from the Antarctic continent → We have added the sentence: "Note that NEWS2 excludes runoff from the Antarctic continent." (L156)

Pg. 5, ln. 133-135: it sounds like this assumes a constant seasonal cycle in runoff; this is not unreasonable in an EMIC where the hydrological cycle may be in a long-term equilibrium → We have added the following text: "Here, we assumed (i) a constant seasonal cycle in runoff and (ii) that nitrogen concentrations in the discharged river water are constant throughout the seasonal cycle. We thus distributed the annual nitrogen load over the months, weighted by the fraction of monthly freshwater discharge." (L166)

Pg. 6, ln. 136: can you clarify what happens with riverine P in this model please?; Figure 1 tends to imply there might be a link between riverine N and P; Global NEWS provides both, and the balance of N and P could be important for your model's N2-fixation response
→ "Riverine phosphorus is not added in this experiment and we generally assume a fixed marine P inventory, like in most previous studies with UVic. The inclusion of a dynamic P cycle (like in Niemeyer et al., 2017, Kemena et al., 2019) with riverine P supply from NEWS2 will be subject to a follow up study". (L169)

Pg. 6, ln. 145: "bioavailable" - maybe add: "(or rapidly turned over to DIN)" → Thanks, this has been added. (L183)

Pg. 7, Table 1: add control experiment to this table → Thank you for this suggestion, CTR has been added (Table 1, Pg. 8)

Pg. 7, ln. 149: "steady state" - it might be nice to see a figure (supplementary?) where the long-term balancing of the N budget took place; for instance, to illustrate the timescales associated with addition and removal processes → Thank you for this suggestion. The timeseries of N is now included as supplement in the revised manuscript (see also Text ln. 189).

Pg. 7, ln. 155: "fairly well" - all runs omit the midwater maximum around 1000 m → "In comparison with observational data of the World Ocean Atlas (Garcia et al., 2019), the model underestimates the observed $NO_3$ in the whole water column by 3 to 4 mmol m$^{-3}$ and especially omits the midwater maximum around 1000 m." (L194)

Pg. 7, ln. 156: "global average" - what is the global concentration difference?; and what is this as a percentage of total observed inventory?
→ The text has been updated to read: "Global average NO3 concentrations only vary a little between the simulations (from 22.19 mmol m-3 in CTR to 22.48 mmol m-3 in NEWS and 22.84 mmol m-3 in 2xDIN) and the differences to CTR correspond to +1.1 %, +1.8 % and +2.5 % of the total observed inventory for NEWS, DIN+DON and 2xDIN respectively. However, the absolute error between model and observations decreases with higher riverine N supply." (L196)

In addition to the numbers in the text and in Table 2 we include some of these numbers here. Note that the differences relative to the observed inventory (Diff / WOA) are very similar to the differences relative to the CTR simulation (fourth column in Table 2 of the manuscript).

| Simulation | Glob. NO3 Conc. [mmol m-3] | NO3 surface | NO3 1100 m | Difference in glob. conc. to CTR | Diff / WOA [%] |
|---|---|---|---|---|---|
| CTR | 22.19 | 5.71 | 24.64 | 0 | |
| NEWS | 22.48 | 5.93 | 24.94 | 0.288 | 1.1 |
| DIN+DON | 22.66 | 6.15 | 25.10 | 0.468 | 1.8 |
| 2xDIN | 22.84 | 6.19 | 25.30 | 0.646 | 2.5 |
| WOA | 26.29 | 6.62 | 31.22 | 4.102 | 15.6 |

Pg. 7, ln. 156: "misfit" - this is absolute error, right? → The text has been changed accordingly (see above) (L200)

Pg. 7, ln. 167: "not surprising" - well, not *entirely* surprising; it's possible, of course, that adding local sources of N might trigger strong balancing denitrification that could even offset the addition (though this seems unlikely) → Yes, we have attenuated this statement as you suggest. (L208)

Pg. 7, ln. 171: on this point, might it be possible to include some total of suboxic ocean volume (e.g. the volume of below some standard oxygen threshold concentration)?
→ We have calculated the global volume of oxygen minimum zone (OMZ) for all simulations, and included the sentence: "Note, that the global volume of ocean minimum zones, defined

here as regions with $O_2$ concentration lower than 70 μmol kg$^{-1}$, is increasing with higher nitrogen supply, from 52 *10$^6$ km$^3$ in CTR, to 54 *10$^6$ km$^3$ in NEWS, 56 *10$^6$ km$^3$ in DIN+DON and 58 *10$^6$ km$^3$ in 2xDIN." (L285)

Pg. 8, Table 2: "+1,12" -> "+1.12" → The typo has been corrected. (Table 2, Pg. 13)

Pg. 10, Figure 5: normally, delta (or bias) plots are coloured blue (negative) to red (positive) with white in the centre –> Thank you for this suggestion, we have replotted all delta figures with a red-white-blue colourbar.

Pg. 11, Figure 6: why 850m?; from Figure 3b, the largest misfit seems to be at 1000m → Figure 6 has been updated with the blue to red colourbar. We chose 850 m and 1100 m depth, because although in the global profiles the largest misfit has been found around 1000 m, at 850 m depth the regions with decrease in No3 concentrations are most pronounced (compare also with Figure 5). (Pg. 11)

Pg. 12, Figure 7: it's obviously not possible to tell how realistic oxygen is here; perhaps compare NEWS and CTR directly to WOA instead? → If we would compare NEWS and CTR to WOA, we would find almost no differences between the two plots, because our UVic simulations show a somewhat different pattern and magnitude in $O_2$ especially in the Arabian Sea and the Gulf of Bengal. What we have done to address your comment is to add four plots to the Figure 7, showing the differences between WOA and CTR for $O_2$ and $NO_3$ in the two ocean basins. (Pg. 14)

Pg. 13, ln. 192: surplus "like" → Yes, corrected, thank you. (L234)

Pg. 13, ln. 193: assuming that N inputs fuel corresponding increases in productivity, it may be worth noting in passing how much production is also enhanced by river N in these regions → We haven't calculated total production in the Bay of Bengal but the analog Figure to Figure 9 and 10 for NPP shows, that productivity is enhanced only very locally near the coast of Myanmar (by up to +36 Tg C yr-1). In the rest of the basins, productivity is lower in the simulations with riverine N supply. We will include this Figure in the supplements and add the information in the text. (L353)

Pg. 13, ln. 197: see my earlier remarks about PO4 availability and riverine sources; if riverine P is neglected, this may skews where N2 fixation is favoured; ditto if PO4 is added in strict proportion to DIN → P is not altered by the additional nitrogen from river supply. The effects of riverine P supply and its interaction with the N cycle will be investigated in a separate study.

Pg. 14, Figure 8: maybe it would be better to show CTR N2 fixation in addition to the deltas? → An additional plot has been added to the panel showing $N_2$-fixation from CTR simulation. (Figure 8, Pg. 16)

Pg. 15, ln. 223: "oxygen concentrations even though higher at the surface" - Is this elevated oxygen due to enhanced production?; typically surface oxygen is boring because it equilibrates quickly to saturation values (with ambient temperature)

→ We have added here: "In the Bay of Bengal, oxygen concentrations appear higher at the surface in NEWS than in CTR by around 1.5 mmol m$^{-3}$ at least in the southern part of the Bay, which could be due to enhanced production. However, compared to WOA, the oxygen concentrations are still very low in the upper 800 m in NEWS like in CTR, and they are particularly low in the NEWS simulations in the subsurface waters and the whole deeper basin (Fig. 7)." (L301)

Pg. 15: per my remarks for Figure 7, making the model's relationship with observed oxygen clear might be useful → Thanks for this suggestion, a comparison with O$_2$ from WOA has been added to the panel plot. (Figure 7, Pg.14)

Pg. 18, Table 3: add a column listing the balance at equilibrium?; I make the discrepancy about 0.5 Tg N / y for all of the model experiments; actually, what is this discrepancy?; is it the model just not fully equilibrated? → Yes, there are discrepancies between 0.4 and 0.5 Tg N / yr in the total sums depending on the simulation, although the model is equilibrated. Nevertheless, UVic has some internal variability and N fixation and denitrification, while nearly balancing each other, can fluctuate in the range of the discrepancies you mention. We will add this information after the table: "Note that the global sums from sources and sinks do not exactly add to zero due to internal variability in the modeled N-cycle." (Pg. 18)

Pg. 18, Figure 11: if possible, it might be an idea to include a map of observational estimated production; the total of the models here might be OK, but I think the patterns - particularly in the Indian Ocean - might not be; this is important given the amount of analysis that is focused in this region → As Keller et al. (2012) already mentioned in their analyses of UVic, simulated primary production is higher in the tropical upwelling regions along the equator and the northern Indian ocean than shown in observations like the dataset by Behrenfeld and Falkowski (1997). We refer e.g. to this publication for the spatial distribution of primary production on line 359 : "In comparison with the NPP pattern in observation (e.g. Behrenfeld and Falkowski, 1997), simulated primary production in UVic is higher in the tropical upwelling regions along the equator and the northern Indian ocean, while there is too little productivity on the shelfs (Keller at al., 2012). In the simulations with riverine N supply, NPP is increased in the coastal regions and, at least for the 2xDIN experiment, lower in the tropical upwelling regions than in CTR." (L359)

Pg. 18, Figure 11: surplus colourbar on Figure 11b? → Figure 11 has been updated with a new (blue-red) colour scheme and without the colourbar on panel (b). (Figure 12, Pg. 21)

Pg. 19, Table 4: this is a weird, single-column table → We have included a column with the method by which the information has been derived. The table now appears as: (Table 5, Pg. 22)

| Source | NPP [Pg C yr-1] | Method |
|---|---|---|
| UVic CTR | 54.9 | model |
| UVic NEWS | 55.3 | model |
| UVic DIN+DON | 55.5 | model |
| UVic 2xDIN | 55.7 | model |

| Behrenfeld and Falkowski (1997) | 43.5 | Satellite data |
|---|---|---|
| Behrenfeld et al. (2005) | 67 | Satellite data |
| Carr et al. (2006) | 51 | Mean of 31 global models |
| Westberry et al. (2008) | 52 | Carbon based, spectral |
| Buitenhuis et al. (2013) | 56 | Model and observational database |

Pg. 19, Table 4: these numbers are a bit higher compared to what I'm used to; e.g. the Oregon State University primary production website; there, I find ~40 Pg C / y; the 3 models their site includes are quite divergent, however
→ Buitenhuis et al (2013) shows an overview from 14 different approaches to calculate global marine primary production rates and the recent one's range between 40 and 60 Pg C / y. From the studies cited by Buitenhuis, the global production as calculated by Westberry et al. (2008) and Behrenfeld et al. (2005) are also in Table 5. We have also analysed the dataset from Behrenfeld and Falkowski (1997), which gives a rate of 43.5 Pg C yr$^{-1}$. To be more complete, we will also include this study in Table 5 (see answer below). (p. 22)

Pg. 20, Table 5: this is a very strange way to organise a table; it's a single column when it should be three columns of numbers→ Yes, sorry for this.  The table has been reorganized in the revised manuscript as you suggest, including supplementary numbers (see responses below). (now Table 4, Pg. 22 and Table 6 Pg. 23)

Pg. 20, Table 5: also, I might be inclined to include numbers (or deltas) for the other major N-cycle processes, N2-fixation and denitrification → Table 5 was originally only a supplement for Figure 11. We had not extracted all the minima and maxima for the other N-cycle processes but for the revised manuscript, we have calculated the global totals for the regions presented in this table and will include them as suggested. (Table 6, Pg. 23)

Pg. 20, Table 5: also, why the areal units here?; would it not make more sense to report global totals (i.e. Pg C / y)? → Table 5 was originally only a supplement for Figure 11 (now Figure 12), to show the extremes in vertically integrated primary production rates. For the revised manuscript we include the regional totals in Tg C / yr for the Bay of Bengal, the Yellow Sea, the North Sea, the outflow of the Rio de la Plata and the East Mediterranean Sea to put the numbers into relation.

Pg. 24, ln. 352: "increase in marine primary production is small" - this analysis appears not to factor in that total riverine input of N is ~0.2% of the N used in primary production; so, contrary to the point you make here, the changes in NPP found between the simulations appear to actually be quite large; I guess the factor that makes the rivers more important is that they deliver N to shallow ocean areas (= shelves) where they will have a larger impact
→ Yes, thanks for your remark. We have added to our text: "The absolute increase in marine primary production is small (between +0.7 % in NEWS and +1.3 % in 2xDIN). However, relative to the amount of N added to the global ocean, primary production increases yearly by 17.5 Tg C per additional Tg N in NEWS (16.0 Tg C per Tg N in 2xDIN). As we have shown, primary production increases mainly near the river mouths, where high nutrient loads are injected in shallow ocean areas, creating production "hot spots", while only small changes in

production have been found in the open ocean. Other studies with additional N supply also found only moderate increase in global primary production rates…" (L440)

Pg. 25, ln. 387: "carbon export" - how is this defined?; e.g. 100 m export?; and does it include or exclude shelf regions where material is not properly exported? → We have added: "Indeed, simulated carbon export, evaluated at the 122.5 m level and including all the shelf regions, increases globally by only 0.06 Pg C / yr in our NEWS simulation." (L485)

Pg. 25, ln. 387: You might want to consider these changes in light of how much nutrient is being added to the ocean relative to N cycling through production; if my earlier back-of-the-envelope calculations are right, river N is 0.2% of N-cycling through production, but you're finding several percent in NPP change
→ We modify our conclusion to: "In the coastal regions and especially in some hot-spot regions near the river mouths, riverine nitrogen input leads to higher primary production. Globally, NPP rates increase up to 17.5 Tg C/yr per Tg N/yr added to the ocean." (L472)

Pg. 25, ln. 390-392: this paragraph doesn't really say very much; I'd suggest deleting it
→ We have shortened it to: ". We have found, that likewise to atmospheric deposition, river supply of nitrogen is not only relevant for the coastal system but also for marine biology in the global ocean. But while atmospheric deposition provides only N, …" (L491)

Pg. 26, ln. 397: if riverine phosphorus is not included here, does that not skew the model's balance between N2-fixation and denitrification since the N being added is not balanced by P? → Yes, this may be true, which is why we will investigate P and N additions in a following study (L496)

Pg. 26, ln. 400: "hdl" - what is this?; could this be put onto Zenodo or something to get a proper DOI? → We apologize for the typo in the link. We have changed it in the revised manuscript it to the thredds link (L499):
https://data.geomar.de/thredds/20.500.12085/59977a36-e8e7-4348-a4e8-2b13f3913590/catalog.html
Thank you for the suggestion of Zenodo.

Pg. 26, ln. 402: I get a 404 error from this link → I tested the link again and it worked for me. Perhaps we could just include the "Home" link which is http://icr.ioc-unesco.org/? In the index on the left it is also possible to find the link to the "Global NEWS".

**Style points:**
Pg. 8, Figure 3: can you make the lines in the key thicker so it's easier to tell them apart?; also, the choice of colours is rather unhelpful in this regard; also, why are they ordered in this strange way in the key? → Thanks for your suggestions. We have changed the lines and the colours of igure 3.

Pg. 9, Figure 4: this colour scale looks more like a delta one to me → We have changed the colours of Figure 4.

Pg. 10, Figure 5: it might be better not to stretch the smaller Indian basin to the same size as the other basins here; perhaps just plot the same latitude range on all three panels? → As this would mean a lot of work to change the range of all Figures with the ocean basins, we have let the plot of the Indian Ocean like it is. But we have included a note, indicating that the ranges are different. (Figure 5, Pg. 11)

Pg. 15, Figure 9: this colourbar implies negative denitification is possible in the model; is it? → We have changed the colourbars of this Figure, using a delta colourbar (blue-red) for panels b-d and have rearranged the colourbar for panel (a) in order to avoid this impression. There are no negative denitrification rates in (a). (Figure 10, Pg. 19)

Pg. 16, Figure 10: this colourbar is missing the extreme cyan colour that indicates "out of range" delta concentration → Thanks for you remark. The colourbar has been changed to blue-red and the cyan colour is now shown correctly. (Figure 11, Pg. 20)

Pg. 23, Figure 13: could you try a clever log scale here?; this plot is otherwise not very Informative → Unfortunately we haven't found another way to plot these results.

Pg. 25, ln. 369: purely as a style point, I would suggest thinning your conclusions section to 5 or 6 bullet-point statements of your findings; this makes it very easy for readers to understand the main findings (and decide whether to read more!) → We have followed your suggestion for the revised manuscript. (L465)
* * *
**Additional references**

Behrenfeld M. J. and Falkowski P. G.: Photosynthesis rates derived from satellite-based chlorophyll concentration, Limnology and Oceanography, 42, 1-20, 1997.

Capone D. G., Zehr J. P., Paerl H. W., Bergman B., and Carpenter E. J.: Trichodesmium, a Globally Significant Marine Cyanobacterium, Science, 276, 1221-1229, 23 May 1997. Cornell S., Rendell A., and Jickells T.: Atmospheric inputs of dissolved organic nitrogen to the oceans, Nature, 376, 243-246, 20 July 1995.

Galbraith, E. D., Gnanadesikan, A., Dunne, J. P., and Hiscock, M. R.: Regional impacts of iron-light colimitation in a global biogeochemical model, Biogeosciences, 7, 1043–1064, doi:10.5194/bg-7-1043-2010, 2010

Kim I.-N, Lee K., Gruber N., Karl D. M., Bullister J. L., Yang S., and Kim T.-W.: Increasing anthropogenic nitrogen in the North Pacific Ocean, Science, 346, Issue 6213, 1102-1105, 28 November 2014.

Krishnamurthy A., Moore J. K., Zender C. S., and Luo C.: Effects of atmospheric inorganic nitrogen deposition on ocean biogeochemistry, J. Geophys. Res., 112, G02019, doi:10.1029/2006JG000334, 2007.

Lamarque J. F., Dentener F., McConell J., Ro C.-U., Shaw M. et al.: Multi-model mean nitrogen and sulfur deposition from the Atmospheric Chemistry and Climate Model Intercomparison Project (ACCMIP): evaluation of historical and projected future changes, Atmos. Chem. Phys., 13, 7997–8018, 2013.

Oschlies A., Koeve W., Landolfi A., and Kähler P., Loss of fixed nitrogen causes net oxygen gain in a warmer future ocean, nature communications, 10:2805, doi.org/10.1038/s41467-019-10813-w, 2019.

Sohm J. A., Webb E. A., and Capone D. G.: Emerging patterns of marine nitrogen fixation, Nature Reviews Microbiology, 9, 499-508, 2011.

Somes, C. J., Schmittner, A., and Altabet, M. A.: Nitrogen isotope simulations show the importance of atmospheric iron deposition for nitrogen fixation across the Pacific Ocean, Geophys. Res. Lett., 37, L23605, doi:10.1029/2010GL044537, 2010a.